# Fast Matrix Square Roots with Applications to Gaussian Processes and Bayesian Optimization

**Geoff Pleiss**
Columbia University
gmp2162@columbia.edu

**Martin Jankowiak**
The Broad Institute
mjankowi@broadinstitute.org

**David Eriksson**[*]
Facebook
deriksson@fb.com

**Anil Damle**
Cornell University
damle@cornell.edu

**Jacob R. Gardner**
University of Pennsylvania
jacobrg@seas.upenn.edu

## Abstract

Matrix square roots and their inverses arise frequently in machine learning, e.g., when sampling from high-dimensional Gaussians $\mathcal{N}(\mathbf{0}, \mathbf{K})$ or "whitening" a vector $\mathbf{b}$ against covariance matrix $\mathbf{K}$. While existing methods typically require $\mathcal{O}(N^3)$ computation, we introduce a highly-efficient quadratic-time algorithm for computing $\mathbf{K}^{1/2}\mathbf{b}$, $\mathbf{K}^{-1/2}\mathbf{b}$, and their derivatives through *matrix-vector multiplication* (MVMs). Our method combines Krylov subspace methods with a rational approximation and typically achieves 4 decimal places of accuracy with fewer than 100 MVMs. Moreover, the backward pass requires little additional computation. We demonstrate our method's applicability on matrices as large as $50,000 \times 50,000$—well beyond traditional methods—with little approximation error. Applying this increased scalability to variational Gaussian processes, Bayesian optimization, and Gibbs sampling results in more powerful models with higher accuracy. In particular, we perform variational GP inference with up to 10,000 inducing points and perform Gibbs sampling on a 25,000-dimensional problem.

## 1 Introduction

High-dimensional Gaussian distributions arise frequently in machine learning, especially in the context of Bayesian modeling. For example, the prior of Gaussian process models is given by a multivariate Gaussian distribution $\mathcal{N}(\mathbf{0}, \mathbf{K})$ governed by an $N \times N$ symmetric positive definite kernel matrix $\mathbf{K}$. Historically, $\mathcal{O}(N^3)$ computation and $\mathcal{O}(N^2)$ memory requirements have limited the tractability of inference for high-dimensional Gaussian latent variable models.

A growing line of research aims to reformulate many common covariance matrix operations—such as linear solves and log determinants—as iterative optimizations involving *matrix-vector multiplications* (MVMs) [e.g. 3, 11, 16, 29, 79]. MVM approaches have two primary advantages: 1) the covariance matrix need not be explicitly instantiated (so only $\mathcal{O}(N)$ memory is required) [11, 16, 79]; and 2) MVMs utilize GPU acceleration better than direct methods like Cholesky [3, 29]. Thus MVM methods can be scaled to much larger covariance matrices.

In this paper, we propose an MVM method that addresses a common computational bottleneck for high-dimensional Gaussians: computing $\mathbf{K}^{\pm 1/2}\mathbf{b}$. This operation occurs frequently in Gaussian process models and inverse problems. For example, if $\mathbf{b} \sim \mathcal{N}(\mathbf{0}, \mathbf{I})$, then $\mathbf{K}^{\frac{1}{2}}\mathbf{b} \sim \mathcal{N}(\mathbf{0}, \mathbf{K})$. This operation appears frequently in Bayesian optimization [e.g. 23, 42, 74, 80] and Gibbs sampling [e.g.

---

[*]This work was conducted while David Eriksson was at Uber AI.

6, 8, 33]. $\mathbf{K}^{-\frac{1}{2}}\mathbf{b}$ can be used to project parameters into a "whitened" coordinate space [50, 54]—a transformation that accelerates the convergence of variational Gaussian process approximations. To make these computations more efficient and scalable, we make the following contributions:

- We introduce a MVM approach for computing $\mathbf{K}^{\pm 1/2}\mathbf{b}$. The approach uses an insight from Hale et al. [35] that expresses the matrix square root as a sum of shifted matrix inverses.

- To efficiently compute these shifted inverses, we leverage a modified version of the MINRES algorithm [59] that performs *multiple shifted solves* through a single iteration of MVMs. We demonstrate that, surprisingly, **multi-shift MINRES (msMINRES)** convergence can be accelerated with a *single* preconditioner despite the presence of multiple shifts. Moreover, msMINRES only requires $\mathcal{O}(N)$ storage when used in conjunction with partitioned MVMs [11, 79]. Achieving 4 or 5 decimal places of accuracy typically requires *fewer than 100 matrix-vector multiplications*, which can be highly accelerated through GPUs.

- We derive a scalable backward pass for $\mathbf{K}^{\pm 1/2}\mathbf{b}$ that enables our approach to be used as part of learning and optimization.

- We apply our $\mathbf{K}^{-1/2}\mathbf{b}$ and $\mathbf{K}^{1/2}\mathbf{b}$ routines to three applications: 1) variational Gaussian processes with up to $M = 10^4$ inducing points (where we additionally introduce a $\mathcal{O}(M^2)$ MVM-based natural gradient update); 2) sampling from Gaussian process posteriors in the context of Bayesian optimization with up to 50,000 candidate points; and 3) an image reconstruction task where we perform Gibbs sampling in 25,600 dimensions.

Code examples for the GPyTorch framework are available at `bit.ly/ciq_variational` and `bit.ly/ciq_sampling`.

## 2  Background

**Existing Methods for Sampling and Whitening** typically rely on the Cholesky factorization: $\mathbf{K} = \mathbf{L}\mathbf{L}^\top$, where $\mathbf{L}$ is lower triangular. Though $\mathbf{L}$ is not a square root of $\mathbf{K}$, $\mathbf{L}\mathbf{b}$ is equivalent to $\mathbf{K}^{1/2}\mathbf{b}$ up to an orthonormal rotation. Therefore, $\mathbf{L}\boldsymbol{\epsilon}$, $\boldsymbol{\epsilon} \sim \mathcal{N}(\mathbf{0}, \mathbf{I})$ can be used to draw samples from from $\mathcal{N}(\mathbf{0}, \mathbf{K})$ and $\mathbf{L}^{-1}\mathbf{b}$ can be used to "whiten" the vector $\mathbf{b}$. However, the Cholesky factor requires $\mathcal{O}(N^3)$ computation and $\mathcal{O}(N^2)$ memory for an $N \times N$ covariance matrix $\mathbf{K}$. To avoid this large complexity, randomized algorithms [58, 63], low-rank/sparse approximations [41, 61, 83], or alternative distributions [80] are often used to approximate the sampling and whitening operations.

**Krylov Subspace Methods** are a family of iterative algorithms for computing functions of matrices applied to vectors $f(\mathbf{K})\mathbf{b}$ [e.g. 65, 67, 77]. Crucially, $\mathbf{K}$ is only accessed through *matrix-vector multiplication* (MVM), which is beneficial for extremely large matrices that cannot be explicitly computed in memory. All Krylov algorithms share the same basic structure: each iteration $j$ produces an estimate $\mathbf{c}_j \approx f(\mathbf{K})\mathbf{b}$ which falls within the $j^{\text{th}}$ *Krylov subspace* of $\mathbf{K}$ and $\mathbf{b}$:

$$\mathbf{c}_j \in \mathcal{K}_j(\mathbf{K}, \mathbf{b}) = \text{span}\left\{\mathbf{b},\ \mathbf{K}\mathbf{b},\ \mathbf{K}^2\mathbf{b},\ \ldots,\ \mathbf{K}^{j-1}\mathbf{b}\right\}. \tag{1}$$

Each iteration expands the Krylov subspace by one vector, requiring a single matrix-vector multiplication with $\mathbf{K}$. Many Krylov methods, such as linear conjugate gradients, can be reduced to computationally efficient vector recurrences. Krylov methods are exact after $N$ iterations, though most methods offer extremely accurate solutions in $J \ll N$ iterations. There has been growing interest in applying Krylov methods to large-scale kernel methods [3, 4, 11, 13, 15, 16, 29, 30, 57, 61, 65, 70, 71, 82], especially due to their memory efficiency and amenability to GPU acceleration.

## 3  Contour Integral Quadrature (CIQ) via Matrix-Vector Multiplication

In this section we develop an MVM method to compute $\mathbf{K}^{-1/2}\mathbf{b}$ and $\mathbf{K}^{1/2}\mathbf{b}$ for sampling and whitening. Our approach scales better than existing methods (e.g. Cholesky) by: 1) reducing computation from $\mathcal{O}(N^3)$ to $\mathcal{O}(N^2)$; 2) reducing memory from $\mathcal{O}(N^2)$ to $\mathcal{O}(N)$; 3) more effectively using GPU acceleration; and 4) affording an efficient gradient computation.

**Contour Integral Quadrature (CIQ).** A standard result from complex analysis is that $\mathbf{K}^{-1/2}$ can be expressed through Cauchy's integral formula: $\mathbf{K}^{-1/2} = \frac{1}{2\pi i} \oint_\Gamma \tau^{-1/2} (\tau \mathbf{I} - \mathbf{K})^{-1} \, \mathrm{d}\tau$, where $\Gamma$

is a closed contour in the complex plane that winds once around the spectrum of $\mathbf{K}$ [18, 35, 44]. Applying a numerical quadrature scheme to the contour integral yields the rational approximations

$$\mathbf{K}^{-\frac{1}{2}} \approx \sum_{q=1}^{Q} w_q \left(t_q \mathbf{I} + \mathbf{K}\right)^{-1} \quad \text{and} \quad \mathbf{K}^{\frac{1}{2}} \approx \mathbf{K} \sum_{q=1}^{Q} w_q \left(t_q \mathbf{I} + \mathbf{K}\right)^{-1}, \tag{2}$$

where the weights $w_q$ encapsulate the normalizing constant, quadrature weights, and the $t_q^{-\frac{1}{2}}$ terms. Hale et al. [35] introduce a real-valued quadrature strategy based on a change-of-variables formulation (described in Appx. B) that converges extremely rapidly—often achieving full machine precision with only $Q \approx 20$ quadrature points. For the remainder of this paper, applying Eq. (2) to compute $\mathbf{K}^{\pm 1/2}\mathbf{b}$ will be referred to as **Contour Integral Quadrature (CIQ)**.

## 3.1  An Efficient Matrix-Vector Multiplication Approach to CIQ with msMINRES.

Using the quadrature method of Eq. (2) for whitening and sampling requires solving several shifted linear systems. To compute the shifted solves required by Eq. (2) we leverage a variant of the minimum residuals algorithm (MINRES) developed by Paige and Saunders [59]. At step $j$ MINRES approximates $\mathbf{K}^{-1}\mathbf{b}$ by the vector within the Krylov subspace $\mathbf{c} \in \mathcal{K}_j(\mathbf{K}, \mathbf{b})$ that minimizes the residual $\|\mathbf{Kc} - \mathbf{b}\|_2$.

**msMINRES for multiple shifted solves.** To efficiently compute all the shifted solves, we leverage techniques [e.g. 14, 17, 24, 25, 55] that exploit the shift-invariance property of Krylov subspaces: i.e. $\mathcal{K}_J(\mathbf{K}, \mathbf{b}) = \mathcal{K}_J(t\mathbf{I}+\mathbf{K}, \mathbf{b})$. We introduce a variant to MINRES, which we refer to as **multi-shift MINRES** or **msMINRES**, that re-uses the same Krylov subspace vectors $[\mathbf{b}, \mathbf{Kb}, \ldots, \mathbf{K}^{J-1}\mathbf{b}]$ for all shifted solves $(t\mathbf{I} + \mathbf{K})^{-1}\mathbf{b}$. In other words, using msMINRES we can get all $(t_q\mathbf{I} + \mathbf{K})^{-1}\mathbf{b}$ *essentially for free*, i.e. only requiring $J$ MVMs for the Krylov subspace $\mathcal{K}_J(\mathbf{K}, \mathbf{b})$. As with standard MINRES, the msMINRES procedure for computing $(t_q\mathbf{I} + \mathbf{K})^{-1}$ from $[\mathbf{b}, \mathbf{Kb}, \ldots, \mathbf{K}^{J-1}\mathbf{b}]$ can be reduced to a simple vector recurrence (see Appx. C for details).

## 3.2  Computational Complexity and Convergence Analysis of msMINRES-CIQ

Pairing Eq. (2) with msMINRES is an efficient algorithm for computing $\mathbf{K}^{1/2}\mathbf{b}$ and $\mathbf{K}^{-1/2}\mathbf{b}$. Alg. 1 (see Appendix) summarizes this approach; below we highlight its computational properties:

**Property 1** (Computation/Memory of msMINRES-CIQ). *$J$ iterations of msMINRES requires exactly $J$ MVMs with the input matrix $\mathbf{K}$, regardless of the number of quadrature points $Q$. The resulting runtime of msMINRES-CIQ is $\mathcal{O}(J\xi(\mathbf{K}))$, where $\xi(\mathbf{K})$ is the time to perform an MVM with $\mathbf{K}$. The memory requirement is $\mathcal{O}(QN)$ in addition to what is required to store $\mathbf{K}$.*

For arbitrary positive semi-definite $N \times N$ matrices, the runtime of msMINRES-CIQ is $\mathcal{O}(JN^2)$, where often $J \ll N$. Performing the MVMs in a map-reduce fashion [11, 79] avoids explicitly forming $\mathbf{K}$, which results in $\mathcal{O}(QN)$ total memory. This is in contrast to Cholesky, which produces an artifact that requires $\mathcal{O}(N^2)$ memory. Below we bound the error of msMINRES-CIQ:

**Theorem 1.** *Let $\mathbf{K} \succ 0$ and $\mathbf{b}$ be inputs to msMINRES-CIQ, producing $\mathbf{a}_J \approx \mathbf{K}^{1/2}\mathbf{b}$ after $J$ iterations with $Q$ quadrature points. The difference between $\mathbf{a}_J$ and $\mathbf{K}^{1/2}\mathbf{b}$ is bounded by:*

$$\left\|\mathbf{a}_J - \mathbf{K}^{\frac{1}{2}}\mathbf{b}\right\|_2 \leq \overbrace{\mathcal{O}\left(\exp\left(-\frac{2Q\pi^2}{\log\kappa(\mathbf{K})+3}\right)\right)}^{\text{Quadrature error}} + \overbrace{\frac{2Q\log\left(5\sqrt{\kappa(\mathbf{K})}\right)\kappa(\mathbf{K})\sqrt{\lambda_{min}}}{\pi}\left(\frac{\sqrt{\kappa(\mathbf{K})}-1}{\sqrt{\kappa(\mathbf{K})}+1}\right)^{J-1}\|\mathbf{b}\|_2}^{\text{msMINRES error}}.$$

*where $\lambda_{max}, \lambda_{min}$ are the max and min eigenvalues of $\mathbf{K}$, and $\kappa(\mathbf{K}) \equiv \frac{\lambda_{max}}{\lambda_{min}}$ is the condition number.*

For $\mathbf{a}'_J \approx \mathbf{K}^{-1/2}\mathbf{b}$, the bound incurs an additional factor of $1/\lambda_{\min}$. (See Appx. G for proofs.) Thm. 1 suggests that error in computing $(t_q\mathbf{I} + \mathbf{K})^{-1}\mathbf{b}$ will be the primary source of error as the quadrature error decays rapidly with $Q$. In many of our applications the rapid convergence of Krylov subspace methods for linear solves is well established, allowing for accurate solutions if desired. For covariance matrices up to $N = 50,000$, often $Q = 8$ and $J \leq 100$ suffices for 4 decimal places of accuracy and $J$ can be further reduced with preconditioning (see Sec. 4 and Appx. D).

### 3.3 Efficient Vector-Jacobi Products for Backpropagation

In certain applications, such as variational Gaussian process inference, we have to compute gradients of the $\mathbf{K}^{-1/2}\mathbf{b}$ operation. This requires the vector-Jacobian product $\mathbf{v}^\top (\partial \mathbf{K}^{-1/2}\mathbf{b}/\partial\mathbf{K})$, where $\mathbf{v}$ is the back-propagated gradient. The form of the Jacobian is the solution to a Lyapunov equation, which requires expensive iterative methods or solving a $N^2 \times N^2$ Kronecker sum $(\mathbf{K}^{1/2} \oplus \mathbf{K}^{1/2})^{-1}$. Both of these options are much slower than the forward pass and are impractical for large $N$. Fortunately, our quadrature formulation affords a computationally efficient approximation to this vector-Jacobian product. If we back-propagate directly through each term in Eq. (2), we have

$$\mathbf{v}^\top \left( \frac{\partial \mathbf{K}^{-1/2}\mathbf{b}}{\partial\mathbf{K}} \right) \approx -\frac{1}{2}\sum_{q=1}^{Q} w_q \left( t_q\mathbf{I} + \mathbf{K} \right)^{-1} \left( \mathbf{v}\mathbf{b}^\top + \mathbf{b}\mathbf{v}^\top \right) \left( t_q\mathbf{I} + \mathbf{K} \right)^{-1}. \qquad (3)$$

Since the forward pass computes the solves with $\mathbf{b}$, the only additional work needed for the backward pass is computing the shifted solves $(t_q\mathbf{I} + \mathbf{K})^{-1}\mathbf{v}$, which can be computed with another call to the msMINRES algorithm. Thus the backward pass takes only $\mathcal{O}(J\xi(\mathbf{K}))$ (e.g. $\mathcal{O}(JN^2)$) time.

### 3.4 Preconditioning

Preconditioners are commonly applied to Krylov subspace methods like MINRES to improve the condition number $\kappa(\mathbf{K})$ and accelerate convergence. However, standard preconditioning techniques do not apply to msMINRES, as each shifted system $\mathbf{K} + t_q\mathbf{I}$ requires its own preconditioner (see Appx. D for details). Each separately preconditioned system would require separate MVMs, defeating the efficiency of msMINRES. Nevertheless, we can use a single preconditioner to compute rotationally-equivalent solutions to $\mathbf{K}^{\pm 1/2}\mathbf{b}$. If $\mathbf{P} \approx \mathbf{K}$ is a preconditioner matrix, we note that:

$$\mathbf{K}\mathbf{P}^{-\frac{1}{2}}(\mathbf{P}^{-\frac{1}{2}}\mathbf{K}\mathbf{P}^{-\frac{1}{2}})^{-\frac{1}{2}}\mathbf{b}, \qquad \mathbf{P}^{-\frac{1}{2}}(\mathbf{P}^{-\frac{1}{2}}\mathbf{K}\mathbf{P}^{-\frac{1}{2}})^{-\frac{1}{2}}\mathbf{b}$$

are equivalent to $\mathbf{K}^{1/2}\mathbf{b}$ and $\mathbf{K}^{-1/2}\mathbf{b}$ (respectively) up to an orthonormal rotation (see Appx. D). We can use msMINRES-CIQ to compute the $(\mathbf{P}^{-1/2}\mathbf{K}\mathbf{P}^{-1/2})^{-1/2}\mathbf{b}$ terms. Crucially, the convergence now depends on the condition number of $\mathbf{P}^{-1/2}\mathbf{K}\mathbf{P}^{-1/2}$, rather than that of $\mathbf{K}$.

### 3.5 Related Work

Other Krylov methods for $\mathbf{K}^{1/2}\mathbf{b}$ and $\mathbf{K}^{-1/2}\mathbf{b}$, often via polynomial approximations [e.g. 44], have been explored. Chow and Saad [13] compute $\mathbf{K}^{1/2}\mathbf{b}$ via a preconditioned Lanczos algorithm. Unlike msMINRES, however, they require storage of the entire Krylov subspace. Moreover this approach does not afford a simple gradient computation. Frommer et al. [26, 27] apply a similar Krylov/quadrature approach to a broad class of matrix functions. More similar to our work is [3, 4], which uses the quadrature formulation of Eq. (2) in conjunction with a shifted conjugate gradients solver. We expand upon their method by: 1) introducing a simple gradient computation; 2) proving a convergence guarantee; and 3) enabling the use of simple preconditioners (see Appx. D).

## 4 Benchmarking msMINRES-CIQ

In this section we empirically measure the convergence and speedup of msMINRES-CIQ applied to several types of covariance matrices.

**Convergence of msMINRES-CIQ.** In Fig. 1 we measure the relative error of computing $\mathbf{K}^{1/2}\mathbf{b}$ with msMINRES-CIQ on random matrices.[2] We vary 1) the number of quadrature points $Q$; 2) the size of the matrix $N$; and 3) the conditioning of the matrix. The left and middle plots display results for matrices with spectra that decay as $\lambda_t = 1/\sqrt{t}$ and $\lambda_t = 1/t^2$, respectively. The right plot displays results for one-dimensional Matérn kernel matrices (formed with random data), which have near-exponentially decaying spectra. Consequently, the $1/\sqrt{t}$ matrices are relatively well-conditioned, while the Matérn kernels are relatively ill-conditioned. Nevertheless, in all cases CIQ achieves $10^{-4}$ relative error with only $Q = 8$ quadrature points, regardless of the size of the matrix.

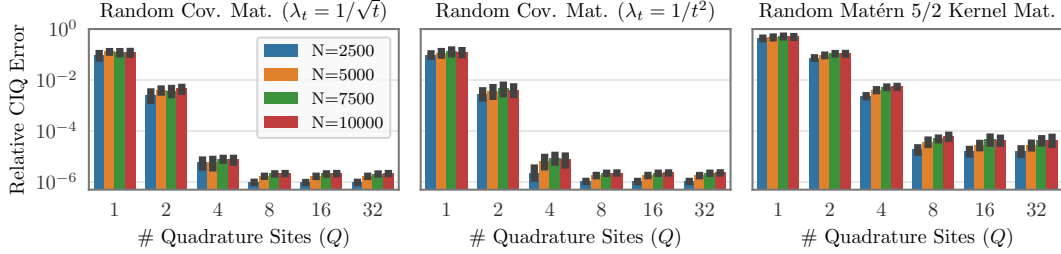

Figure 1: msMINRES-CIQ relative error when computing $\mathbf{K}^{1/2}\mathbf{b}$ as a function of number of quadrature sites $Q$. We test random matrices with eigenvalues that scale as $\lambda_t = 1/\sqrt{t}$ (left) and $\lambda_t = 1/t^2$ (middle), as well as Matérn kernels (right). In all cases $Q = 8$ achieves $< 10^{-4}$ error. The error levels out at roughly $10^{-4}$ or $10^{-5}$, which corresponds to the msMINRES tolerance. msMINRES is stopped after achieving a relative residual of $10^{-4}$ or $J = 400$ iterations.

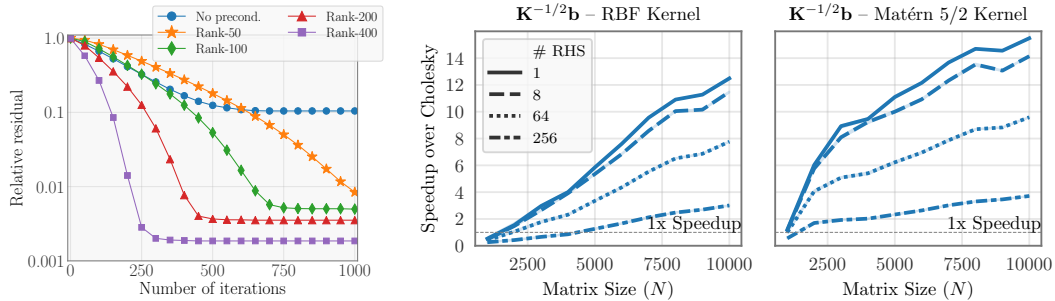

Figure 2: (**Left**:) Effect of preconditioning on msMINRES-CIQ convergence while performing Bayesian optimization. Samples are drawn from the $N = 50,000$ posterior covariance matrix of the ill-conditioned 6-dimensional Hartmann function (see Sec. 5.2), using the pivoted Cholesky preconditioner [29]. (**Middle/Right**:) Speedup of msMINRES-CIQ over Cholesky when computing forward/backward passes of $\mathbf{K}^{-1/2}\mathbf{b}$ with varying number of right-hand-sides $\mathbf{b}$ (RHS).

Additionally, Appx. A demonstrates that msMINRES-CIQ achieves orders of magnitude smaller error than approximation algorithms like randomized SVD [36] or random Fourier features [63].

To demonstrate the effect of preconditioning, we construct a posterior covariance matrix of size $N = 50,000$ points on the 6 dimensional Hartmann function (see Sec. 5.2 for a description). We note that this problem is particularly ill-conditioned ($\kappa(\mathbf{K}) \approx 10^8$), and thus represents an extreme test case. Fig. 2 (left) plots the convergence of msMINRES-CIQ (computing $\mathbf{K}^{1/2}\mathbf{b}$). Without preconditioning, it is difficult to achieve relative residuals less than $0.1$. Using the pivoted Cholesky preconditioner of Gardner et al. [29]—a low-rank approximation of $\mathbf{K}$—not only accelerates the convergence but also reduces the final residual. With rank-200/rank-400 preconditioners, the final residual is cut by orders of magnitude, and msMINRES-CIQ converges $2\times/4\times$ faster.

**Speedup over Cholesky.** We compare the wall-clock speedup of msMINRES-CIQ over Cholesky in Fig. 2 (middle/right) on RBF/Matérn kernels.[3] We compute $\mathbf{K}^{-1/2}\mathbf{b}$ and its derivative on multiple right-hand-side (RHS) vectors. As $N$ increases, msMINRES-CIQ incurs a larger speedup (up to $15\times$ faster than Cholesky). This speedup is less pronounced when computing many RHSs simultaneously, as the cubic complexity of Cholesky is amortized across each RHS. Nevertheless, msMINRES-CIQ is advantageous for matrices larger than $N = 3,000$ even when simultaneously whitening 256 vectors.

## 5 Applications

In previous sections we showed, both theoretically and empirically, that msMINRES-CIQ accurately computes $\mathbf{K}^{\pm1/2}\mathbf{b}$ while scaling better than traditional (Cholesky-based) methods. In this section we

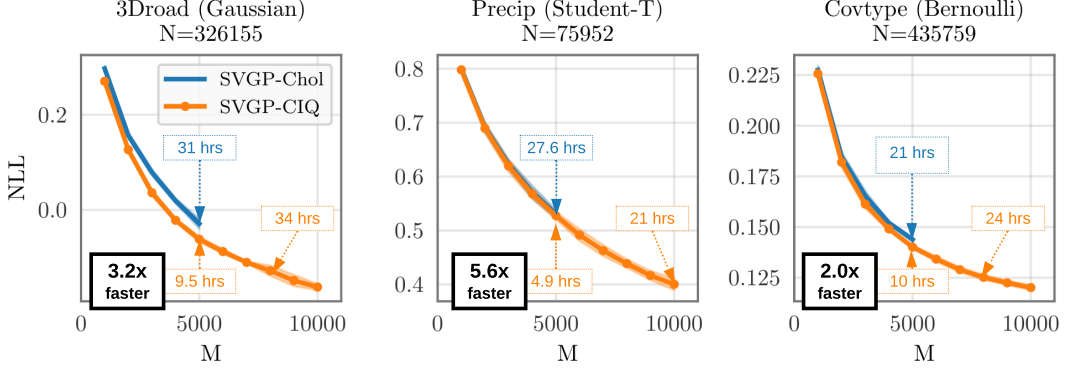

Figure 3: Negative log likelihood (NLL) comparison of Cholesky vs CIQ SVGP models. **Left:** 3DRoad dataset ($N = 326155, D = 2$, Gaussian likelihood). **Middle:** Precipitation dataset ($N = 75952, D = 3$, Student-T likelihood). **Right:** CoverType dataset ($N = 435759, D = 54$, Bernoulli likelihood). NLL improves with more inducing points ($M$), and Cholesky and msMINRES-CIQ models have similar performance. However CIQ models train faster than their Cholesky counterparts.

demonstrate applications of this increased speed and scalability. In particular, we show that using msMINRES-CIQ in conjunction with variational Gaussian processes, Bayesian optimization, and Gibbs sampling facilitates higher-fidelity models that can be applied to large-scale problems.

## 5.1 Whitened Stochastic Variational Gaussian Processes

As a first application, we demonstrate that the msMINRES-CIQ whitening procedure $\mathbf{K}^{-1/2}\mathbf{b}$ can increase the fidelity of **stochastic variational Gaussian processes (SVGP)** [39, 40, 54]. These models are used for non-conjugate likelihoods (e.g. binary classification) or for large datasets that do not fit into memory. SVGP forms an approximate posterior $p(f(\mathbf{x}) \mid \mathbf{X}, \mathbf{y}) \approx q(f(\mathbf{x})) = \mathbb{E}_{q(\mathbf{u})}[p(f(\mathbf{x}) \mid \mathbf{u})]$, where $\mathbf{u} \in \mathbb{R}^M$ are inducing function values (see [40, 54] for a detailed derivation). $q(\mathbf{u})$ is a Gaussian variational distribution parameterized by mean $\mathbf{m} \in \mathbb{R}^M$ and covariance $\mathbf{S} \in \mathbb{R}^{M \times M}$. $\mathbf{m}$ and $\mathbf{S}$ (as well as the model's kernel/likelihood hyperparameters) are chosen to maximize the variational ELBO:

$$\mathcal{L}_{\text{ELBO}}\{q(\mathbf{u}) = \mathcal{N}(\mathbf{m}, \mathbf{S})\} = \sum_{i=1}^{N} \mathbb{E}_{q(f(\mathbf{x}^{(i)}))}\left[\log p(y^{(i)} \mid f(\mathbf{x}^{(i)}))\right] - \text{KL}\left[q(\mathbf{u})\|p(\mathbf{u})\right].$$

Rather than directly learning $\mathbf{m}$ and $\mathbf{S}$, it is more common to learn the *whitened parameters* [50, 54]: $\mathbf{m}' = \mathbf{K}_{\mathbf{ZZ}}^{-1/2}\mathbf{m}$ and $\mathbf{S}' = \mathbf{K}_{\mathbf{ZZ}}^{-1/2}\mathbf{S}\mathbf{K}_{\mathbf{ZZ}}^{-1/2}$. Under these coordinates, the KL divergence term is $\frac{1}{2}(\mathbf{m}'^\top \mathbf{m}' + \text{Tr}(\mathbf{S}') - \log|\mathbf{S}'| - M)$, which doesn't depend on $p(\mathbf{u})$ and therefore is relatively simple to optimize. The posterior distribution $q(f(\mathbf{x})) = \mathcal{N}(\mu_{\text{aprx}}^*(\mathbf{x}), \text{Var}_{\text{aprx}}^*(\mathbf{x}))$ is given by

$$\mu_{\text{aprx}}^*(\mathbf{x}) = \mathbf{k}_{\mathbf{Zx}}^\top \mathbf{K}_{\mathbf{ZZ}}^{-\frac{1}{2}}\mathbf{m}', \quad \text{Var}_{\text{aprx}}^*(\mathbf{x}) = k(\mathbf{x},\mathbf{x}) - \mathbf{k}_{\mathbf{Zx}}^\top \mathbf{K}_{\mathbf{ZZ}}^{-\frac{1}{2}}(\mathbf{I} - \mathbf{S}')\mathbf{K}_{\mathbf{ZZ}}^{-\frac{1}{2}}\mathbf{k}_{\mathbf{Zx}}. \qquad (4)$$

**Time and space complexity.** During training, we repeatedly compute the ELBO and its derivative, which requires computing Eq. (4) and its derivative for a minibatch of data points. Optimization typically requires up to 10,000 iterations of training [e.g. 66]. We note that $\mathbf{K}_{\mathbf{ZZ}}^{-1/2}\mathbf{b}$ (and its derivative) is the most expensive numerical operation during each ELBO computation. If we use Cholesky to compute this operation, the time complexity of SVGP training is $\mathcal{O}(M^3)$.[4] On the other hand, msMINRES-CIQ-based SVGP training is only $\mathcal{O}(JM^2)$, where $J$ is the number of msMINRES iterations. Both methods require $\mathcal{O}(M^2)$ storage for the $\mathbf{m}'$ and $\mathbf{S}'$ parameters.

**Natural gradient descent with msMINRES-CIQ.** The size of the variational parameters $\mathbf{m}'$ and $\mathbf{S}'$ grows quadratically with $M$. This poses a challenging optimization problem for standard gradient descent methods. To adapt to the large $M$ regime, we rely on **natural gradient descent (NGD)** to optimize $\mathbf{m}'$ and $\mathbf{S}'$ [e.g. 38, 66]. At a high level, these methods perform the updates $[\mathbf{m}, \ \mathbf{S}] \leftarrow [\mathbf{m}, \ \mathbf{S}] - \varphi \, \boldsymbol{\mathcal{F}}^{-1} \nabla \mathcal{L}_{\text{ELBO}}$, where $\varphi$ is a step size, $\nabla \mathcal{L}_{\text{ELBO}}$ is the ELBO gradient, and $\boldsymbol{\mathcal{F}}$ is the

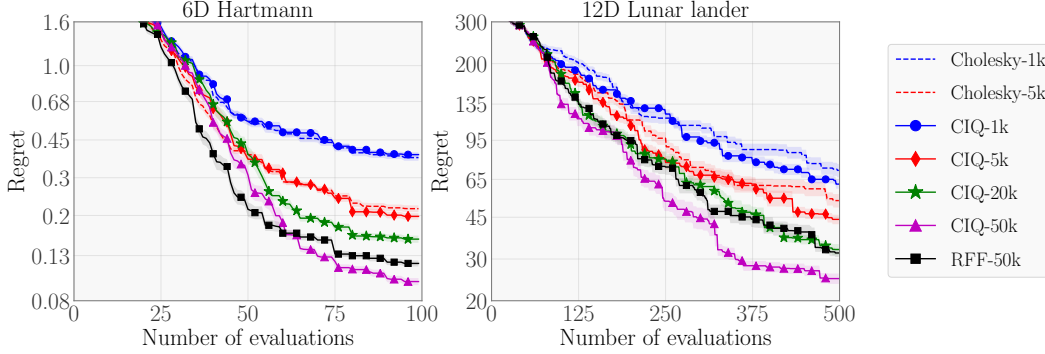

Figure 4: A comparison of sampling methods for Bayesian Optimization. BO is applied to the (**left**) Hartmann ($D = 6$) and (**right**) Lunar Lander ($D = 12$) problems. Methods: Cholesky-$\langle T \rangle$ draws posterior samples with Cholesky at $T$ candidate points. CIQ-$\langle T \rangle$ draws posterior samples with msMINRES-CIQ. RFF-50k uses random Fourier features to draw approximate posterior samples at 50,000 candidate points. Larger $T$ results in better optimization. msMINRES-CIQ enables scaling to $T \geq 50,000$. Each plot shows mean regret with standard error in log-scale based on 30 replications.

Fisher information matrix of the variational parameters. Naïvely, each NGD step requires $\mathcal{O}(M^3)$ computations with $\mathbf{m}'$ and $\mathbf{S}'$, which would dominate the cost of CIQ-based SVGP. Fortunately, we can derive a natural gradient update that only relies on matrix solves with $\mathbf{S}'$, which take $\mathcal{O}(JM^2)$ time using preconditioned conjugate gradients. Therefore, using NGD incurs the same *quadratic* asymptotic complexity as msMINRES-CIQ. See Appx. E for the $\mathcal{O}(M^2)$ NGD update equations.

**Cholesky vs msMINRES-CIQ.** We compare msMINRES-CIQ-SVGP against Cholesky-SVGP on 3 large-scale datasets: a GIS dataset (**3droad**, $D = 2$) [34], a monthly precipitation dataset (**Precipitation**, $D = 3$) [52, 53], and a tree cover dataset (**Covtype**, $D = 54$) [9].[5] Each task has between $N = 70,000$ and $500,000$ training data points. For 3droad we use a Gaussian observation model. The Precipitation dataset has noisier observations; therefore we apply a Student-T observation model. Finally, we reduce the CovType dataset to a binary classification problem and apply a Bernoulli observation model.[6] We train models with $10^3 \leq M \leq 10^4$. See Appx. F for details.

The two methods achieve very similar test-set negative log likelihood (Fig. 3). We note that there are small differences in the optimization dynamics, which is to be expected since $\mathbf{K}_{\mathbf{ZZ}}^{-1/2} \mathbf{k}_{\mathbf{Zx}}$ can differ by an orthogonal transformation when computed with msMINRES-CIQ versus Cholesky. The key difference is the training time: with $M = 5,000$ inducing points, msMINRES-CIQ models are up to *5.6x faster* than Cholesky models (on a Titan RTX GPU). Moreover, msMINRES-CIQ models with $M = 8,000\text{-}10,000$ take roughly the same amount of time as $M = 5,000$ Cholesky models. This speed is due to the rapid convergence of msMINRES—on average $J = 100$ kernel-vector multiplies suffices to achieve 3 decimal places of error (see Appx. A). Note we do not train $M > 5,000$ Cholesky models as doing so would require 14GB of GPU memory and 2-10 days for training.

**Effects of increased inducing points.** We find that accuracy improves with increased $M$ on all datasets. Scaling from $M = 5,000$ to $M = 10,000$ reduces test-set NLL by 0.1 nats on the 3droad and Precipitation datasets. We find similar reductions in predictive error (see Appx. A for plots). By scaling more readily to large $M$, msMINRES-CIQ enables high-fidelity variational approximations that would be computationally prohibitive with Cholesky.

## 5.2 Posterior Sampling for Bayesian Optimization

The second application of msMINRES-CIQ we explore is Gaussian process posterior sampling in the context of Bayesian optimization (BO) [e.g. 72]. Many acquisition functions require drawing samples from posteriors [e.g. 23, 42, 80]. One canonical example is **Thompson Sampling** (TS) [43, 47, 74]. TS trades off exploitation of existing minima for exploration of new potential minima. TS chooses the next query point $\widetilde{\mathbf{x}}$ as the minimizer of a sample drawn from the posterior. Let $\mathbf{X}^* = [\mathbf{x}_1^*, \ldots, \mathbf{x}_T^*]$ be

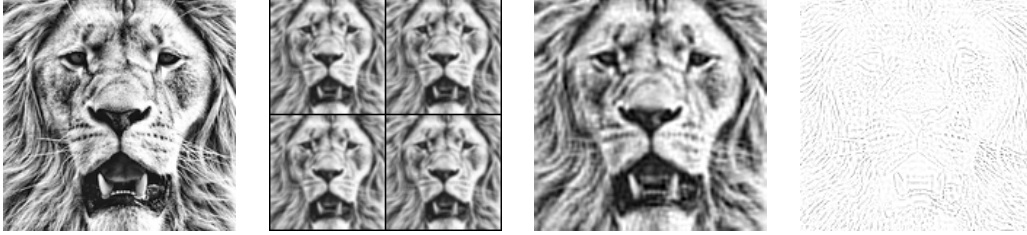

Figure 5: Using msMINRES-CIQ for solving problems in spatial statistics, such as image reconstruction. This requires sampling from a precision matrix of dimension $D = 25{,}600$. (**Left**) High-resolution image of dimension $D$. (**Middle Left**) Low-resolution images. (**Middle Right**) Reconstructed image. (**Right**) Delta between original image and reconstruction (darker colors correspond to larger deltas).

a *candidate set* of possible query points. To choose the next query point $\widetilde{\mathbf{x}}$, TS computes

$$\widetilde{\mathbf{x}} = \arg\min\left(\boldsymbol{\mu}^*(\mathbf{X}^*) + \mathbf{COV}^*(\mathbf{X}^*)^{\frac{1}{2}}\boldsymbol{\epsilon}\right), \quad \boldsymbol{\epsilon} \sim \mathcal{N}(\mathbf{0}, \mathbf{I}). \tag{5}$$

where $\boldsymbol{\mu}^*(\mathbf{X}^*)$ and $\mathbf{COV}^*(\mathbf{X}^*)$ are the posterior mean and covariance of the Gaussian process at the candidate set. The candidate set is often chosen using a space-filling design, e.g. a Sobol sequence. The search space grows exponentially with the dimension; therefore, we need large values of $T$ to more densely cover the search space for better optimization performance. Using Cholesky to compute Eq. (5) incurs a $\mathcal{O}(T^3)$ computational cost and $\mathcal{O}(T^2)$ memory, which severely limits the size of $T$. In comparison, msMINRES-CIQ only requires $\mathcal{O}(T^2)$ computation and $\mathcal{O}(T)$ memory.

We perform BO using TS on the classic test function (**Hartmann**, $D = 6$) and a reinforcement controller tuning problem (**Lunar Lander**, $D = 12$) from the OpenAI gym.[7] We provide more details in the supplementary material. For each problem we use exact Gaussian processes as the surrogate model and TS as the acquisition function. Our goal is to determine whether CIQ-based sampling is beneficial by enabling scaling to larger candidate set sizes.

**Baselines.** We measure the performance of TS as a function of the candidate set size $T$ and consider $T \in \{1{,}000, 5{,}000, 20{,}000, 50{,}000\}$. We run Cholesky (**Cholesky**-$T$) for $T \in \{1{,}000, 5{,}000\}$ and msMINRES-CIQ (**CIQ**-$T$) for $T \geq 5{,}000$. Note that it would be very challenging and impractical to use Cholesky with $T \geq 10{,}000$, due to its quadratic memory and cubic time complexity. For example, running Cholesky for $T = 50{,}000$ would require $\geq 100$ GB of GPU memory, and performing a single decomposition would take (at best) $\approx 30$ seconds. In addition to Cholesky and CIQ with exact Gaussian processes as the surrogate model, we also compare to random Fourier features (RFF) [63] with 1,000 random features.

**Optimization performance.** We plot the mean regret with standard error based on 30 replications in Fig. 4. By increasing $T = 1{,}000$ to $T = 50{,}000$, the final regret achieved by CIQ is significantly lower on both problems. We re-iterate that $T = 50{,}000$ is largely impractical with Cholesky. Large candidate sets have previously only been possible with approximate sampling methods like RFF. We note, however, that RFF with $T = 50{,}000$ is outperformed by CIQ-50k on both problems.

## 5.3 Gibbs Samplers and Image Reconstruction

High-dimensional Gaussian distributions are ubiquitous in Bayesian statistics, especially in the context of spatially structured data. Application areas are numerous, including disease mapping, archaeology, and image analysis [8, 49, 78]. Many of the models that arise in these applications are amenable to Gibbs sampling, a MCMC method for generating (approximate) samples from Bayesian posteriors. As such, sampling from high-dimensional Gaussian distributions is often the primary computational bottleneck for these methods.

To illustrate the utility of msMINRES-CIQ for constructing efficient Gibbs samplers for high-dimensional Gaussian latent variables, we consider an image reconstruction task [6]. We emphasize, however, the wide-ranging applicability of these methods, including for non-spatially structured data

(e.g. for sparse linear regression [33]). We formulate an image analysis model as follows: we observe $R$ low-resolution images $\{\mathbf{y}_r\}_{r=1}^{R}$, with each image of size $M \times M$. The goal is to reconstruct the unknown high-resolution image $\mathbf{x}$ of size $N \times N$ with $N > M$. The joint density is given by

$$p(\mathbf{x}, \mathbf{y}_{1:R}, \gamma_{\text{obs}}, \gamma_{\text{prior}}) = \mathcal{N}(\mathbf{y}_{1:R}|\mathbf{A}\mathbf{x}, \gamma_{\text{obs}}^{-1}\mathbf{1})\mathcal{N}(\mathbf{x}|\mathbf{0}, \gamma_{\text{prior}}^{-1}\mathbf{L})p(\gamma_{\text{obs}})p(\gamma_{\text{prior}}) \qquad (6)$$

where $\mathbf{A}$ is a $M^2R \times N^2$ matrix that encodes how the high-resolution image is blurred and down-sampled to yield $R$ low-resolution images and $\mathbf{L}$ is a $N^2 \times N^2$ discrete Laplace operator that encodes our prior smoothness assumptions about the image $\mathbf{x}$. Additionally, $\gamma_{\text{obs}}$ and $\gamma_{\text{prior}}$ are scalar hyperparameters that control the scale of the observation noise and strength of the image prior, respectively. For more details please refer to Appendix F. The computational bottleneck in the resulting Gibbs sampler is sampling from the conditional Gaussian distribution given by

$$p(\mathbf{x}|\mathbf{y}_{1:R}, \gamma_{\text{obs}}, \gamma_{\text{prior}}) = \mathcal{N}(\mathbf{x}|\mathbf{m}, \mathbf{\Lambda}^{-1}) \quad \mathbf{m} = \gamma_{\text{obs}}\mathbf{\Lambda}^{-1}\mathbf{A}^T\mathbf{y}_{1:R} \quad \mathbf{\Lambda} = \gamma_{\text{obs}}\mathbf{A}^T\mathbf{A} + \gamma_{\text{prior}}\mathbf{L}$$

For a concrete demonstration we perform image reconstruction on the image depicted in Fig. 5. Here $N = 160$, $M = 80$, and $R = 4$, so that the precision matrix $\mathbf{\Lambda}$ is of size $25600 \times 25600$. Despite the extreme size, our implementation achieves $\approx 0.61$ samples per second (using a TitanRTX GPU). We estimate that a Cholesky version of this method would achieve only $\approx 0.05$ samples per second.

## 6  Discussion

We have introduced msMINRES-CIQ—a MVM-based method for computing $\mathbf{K}^{1/2}\mathbf{b}$ and $\mathbf{K}^{-1/2}\mathbf{b}$. In sampling and whitening applications, msMINRES-CIQ can be used as a $\mathcal{O}(N^2)$ drop-in replacement for the $\mathcal{O}(N^3)$ Cholesky decomposition. Its scalability and GPU utilization enable us to use more inducing points with SVGP models and larger candidate sets in Bayesian optimization. In all applications, such increased fidelity results in better performance.

**Stability of msMINRES-CIQ.** Krylov methods on symmetric matrices can be prone to numerical instabilities due to round-off errors [e.g. 60]. Our method has two key advantages that improve stability. First, we only use Krylov methods to solve linear systems rather than eigenvalue problems. Common numerical pitfalls that hinder Krylov eigen-solvers (e.g. loss of orthogonality between Lanczos vectors) have been shown to have little empirical effect on linear system solvers like MINRES and CG [e.g. 22, 75]. Second, each solve from msMINRES is inherently a shifted system $\mathbf{K} + t_q\mathbf{I}$. In practice these shifts dramatically improve the conditioning of $\mathbf{K}$, and allow us to work directly with the matrix $\mathbf{K}$ without having to add diagonal jitter for stability.

**Comparison to other fast sampling methods.** Historically, GP samples have been drawn using the Cholesky factor or finite-basis approximations like RFFs. Recently, a growing line of work investigates using inducing point methods for scalable sampling [61, 83]. We believe that CIQ-sampling can be used in conjunction with these inducing point approaches. For example, Wilson et al. [83] use RFFs to sample from the prior and an inducing point approximation of the conditional to convert prior samples into posterior samples. CIQ can augment this approach, allowing for more inducing points and/or replacing RFFs for prior sampling.

**Advantages and disadvantages.** One advantage of the Cholesky decomposition is its reusability. As discussed in Sec. 4, the cubic cost of computing $\mathbf{L}\mathbf{L}^\top$ is amortized when drawing $\mathcal{O}(M)$ samples or whitening $\mathcal{O}(M)$ vectors. Conversely, applying msMINRES-CIQ to $\mathcal{O}(M)$ vectors would incur a $\mathcal{O}(M^3)$ cost, eroding its computational benefits. Thus, our method is primarily advantageous in scenarios with a small number of right hand sides or where $\mathbf{K}$ is too large to apply Cholesky. We also emphasize that msMINRES-CIQ—like all Krylov methods—can take advantage of fast MVMs afforded by structured covariances. Though this paper focuses on applying this algorithm to dense matrices, we suggest that future work explore applications involving sparse or structured matrices.

## Broader Impact

This paper introduces an algorithm to improve the efficiency and scalability of a common-place computation. The results section highlights three common use cases of this algorithm: variational Gaussian processes, Bayesian optimization, and Gibbs sampling. While there are other potential use-cases of this method, we will focus on the broader impacts with respect to these three applications.

Variational Gaussian processes and Gibbs sampling are common methods. Other researchers have focused on domains like medicine [28, 68], geo-statistics [19, 73], and time-series modelling [64, 81] to motivate the need for increased scalability and efficiency. We believe that our proposed algorithm will make Gaussian process models and Gibbs sampling techniques increasingly applicable in these settings. Researchers/practitioners in these fields might have previously been unable to use Gaussian processes/Gibbs sampling due to scalability issues. While we believe increasing the scalability and usability of these probabilistic techniques is a worthwhile goal, we note that they require additional care when using. If a system is to rely on probabilistic methods for calibrated uncertainty estimates, it will no longer be sufficient to iterate on accuracy as a target method. We also note that performing meaningful probabilistic inferences requires some level of domain expertise regarding modeling priors and potential biases of sampling/variational approximations.

Bayesian optimization is a tool commonly used for hyperparameter optimization [72], A/B testing [5], and other black-box optimization problems. One of the most popular and best performing acquisition functions is Thompson sampling, which requires sampling the unknown function at a candidate set. The primary benefit of the proposed method is better optimization, which could lead to better machine learning models (via better hyperparameter searches) and faster experimental testing (via A/B testing). We would argue that improving the efficiency of such algorithms poses minimal risk beyond more general concerns about potential misapplications of the underlying technology to the optimization of nefarious objectives, intentionally or otherwise. However, we will make note here of some general risks associated with black-box optimization: a potential over-reliance on fully automated methods and computationally expensive searches for what might be marginal improvements.

We have release an open-sourced implementation of this algorithm to facilitate the adoption of this method.[8] Since our method relies on quadrature approximations and iterative refinement, one mode of failure is when such iterations fail to converge to a good estimate (for example, due to bad conditioning). However, there are several easy-to-perform convergence checks (e.g. the msMINRES residual), and such convergence checks are part of our implementation to catch such failure cases.

## Acknowledgments and Disclosure of Funding

We thank David Bindel for helpful conversations about rational approximations and optimization. At the time of submission, GP was support by grants from the National Science Foundation NSF (III-1618134, III-1526012, IIS-1149882, IIS- 1724282, OAC-1934714, and TRIPODS-1740822), the Office of Naval Research DOD (N00014-17-1-2175), the Bill and Melinda Gates Foundation, and the Cornell Center for Materials Research with funding from the NSF MRSEC program (DMR-1719875). AD is partially funded by the National Science Foundation under award DMS-1830274 We are thankful for generous support by Zillow and SAP America Inc.

## Footnotes

[2] msMINRES is stopped after achieving a relative residual of $10^{-4}$ or after reaching $J = 400$ iterations.

[3] $Q = 8$. msMINRES is stopped after a residual of $10^{-4}$. Kernels are formed using data from the Kin40k dataset [2]. Timings are performed on a NVIDIA 1070 GPU.

[4] Note that Cholesky computes $\mathbf{K}^{-1/2}\mathbf{b}$ up to an orthogonal rotation, which is suitable for whitened SVGP.

[5] Details on these datasets (including how to acquire them) are in Appx. F.

[6] The task is predicting whether the primary tree cover at a given location is pine trees or other types of trees.

[7] https://gym.openai.com/envs/LunarLander-v2

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
