[Supplementary Material]

# Supplementary Information for: Fast Matrix Square Roots with Applications to Gaussian Processes and Bayesian Optimization

**Geoff Pleiss**
Columbia University
gmp2162@columbia.edu

**Martin Jankowiak**
The Broad Institute
mjankowi@broadinstitute.org

**David Eriksson**[*]
Facebook
deriksson@fb.com

**Anil Damle**
Cornell University
damle@cornell.edu

**Jacob R. Gardner**
University of Pennsylvania
jacobrg@seas.upenn.edu

---

**Algorithm 1:** Computing $\mathbf{K}^{-\frac{1}{2}}\mathbf{b}$ with MVM-based Contour Integral Quadrature (CIQ)

---

**Input** : `mvm_K` $(\cdot)$ – function for matrix-vector multiplication (MVM) with matrix $\mathbf{K}$
$\qquad$ $\mathbf{b}$ – right hand side, $J$ – number of `msMINRES` iterations, $Q$ – number of quad. points

**Output** : $\mathbf{a} \approx \mathbf{K}^{-\frac{1}{2}}\mathbf{b}$

$[w_1, \ldots, w_Q], [t_1, \ldots, t_Q] \leftarrow$ `compute_quad(` `mvm_K` `(·),` $Q$ `)` // Weights ($w_i$) and
$\quad$ shifts ($t_i$) for quadrature – details in Appx. B.
$(t_1\mathbf{I} + \mathbf{K})^{-1}\mathbf{b}, \ldots (t_Q\mathbf{I} + \mathbf{K})^{-1}\mathbf{b} \leftarrow$ `msMINRES(` `mvm_K` `(·),` $\mathbf{b}$, $J$, $t_1$, ..., $t_Q$ `)`
$\quad$ // msMINRES computes all solves simultaneously – details in
$\quad$ Appx. C.

**return** $\sum_{q=1}^{Q} w_q (t_q\mathbf{I} + \mathbf{K})^{-1}\mathbf{b}$ // CIQ estimate of
$\frac{1}{2\pi i} \int \tau^{-1/2}(\tau\mathbf{I} - \mathbf{K})^{-1}\mathbf{b}\, d\tau = \mathbf{K}^{-1/2}\mathbf{b}$

---

## A  Additional Results

Figure S1: CIQ relative error at computing $\mathbf{K}^{1/2}\mathbf{b}$ as a function of number of quadrature points $Q$. In all cases $Q = 8$ achieves $< 10^{-4}$ error.

Fig. S1 and Fig. S2 are continuations of Fig. 1. They plots CIQ convergence and randomized SVD convergence as a function of $Q$ and $R$ for covariance matrices whose eigenvalues decay as $\lambda_t = \frac{1}{\sqrt{t}}$, $\lambda_t = \frac{1}{t}$, $\lambda_t = \frac{1}{t^2}$, and $\lambda_t = \exp(-t)$ in addition to the kernel matrix results already presented. The

---

[*]This work was conducted while David Eriksson was at Uber AI.

Figure S2: Randomized SVD relative error at computing $\mathbf{K}^{1/2}\mathbf{b}$ as a function of approximation rank $R$. In all cases, randomized SVD is unable to achieve a relative error better than about $0.25$.

results for CIQ demonstrate that it is relatively invariant to the eigenvalue decay speed, and does not require approximately low rank structure. Randomized SVD on the other hand incurs an order of magnitude more error; a rank of $1{,}024$ is unable to reduce the relative error to a single decimal point.

Figure S3: Effect of preconditioning on CIQ convergence (random RBF and Matérn-5/2 kernels with a pivoted Cholesky preconditioner [29]).

Fig. S3 further demonstrates the effect of preconditioning on msMINRES-CIQ. We construct random $N \times N$ RBF/Matérn kernels, applying msMINRES-CIQ to a set of $N$ orthonormal vectors ($[\mathbf{K}^{1/2}\mathbf{b}_1, \ldots, \mathbf{K}^{1/2}\mathbf{b}_N]$), and compute the empirical covariance. We plot the number of msMINRES iterations needed to achieve a relative error of $10^{-4}$. The pivoted Cholesky preconditioner of Gardner et al. [29]—which forms a low-rank approximation of $\mathbf{K}$—accelerates convergence of msMINRES. Without preconditioning (i.e. rank=0), $J = 100$ iterations are required for $N = 7{,}500$ matrices. With rank-100/rank-400 preconditioners, iterations are cut by a factor of two/four.

To further compare msMINRES-CIQ to randomized methods, Fig. S4 plots the empirical covariance matrix of $1{,}000$ Gaussian samples drawn from a Gaussian process prior $\mathcal{N}(\mathbf{0}, \mathbf{K})$. We construct the RBF covariance matrices $\mathbf{K}$ using subsets of the Protein and Kin40k datasets[9] [2]. We note that all methods incur some sampling error, regardless of the subset size ($N$). msMINRES-CIQ and Cholesky-based sampling tend to have very similar empirical covariance error. On the other hand, the

Figure S4: Empirical covariance error (relative norm) for various sampling methods (Cholesky, msMINRES-CIQ, and 1,000 Random Fourier Features [63]). Empirical covariances are measured from 1,000 samples. RBF matrices are constructed from data in the Protein and Kin40k datasets [2].

Random Fourier Features method [63] (with 1,000 random features) incurs errors up to $2\times$ as large. This additional error is due to the randomness in the RFF approximation.

Figure S5: Error comparison of Cholesky-whitened vs CIQ-whitened SVGP models. **Left:** 3DRoad dataset RMSE ($N = 326155$, $D = 2$, Gaussian likelihood). **Middle:** Precipitation dataset RMSE ($N = 75952$, $D = 3$, Student-T likelihood). **Right:** CoverType dataset $0/1$ error ($N = 435759$, $D = 54$, Bernoulli likelihood). Error improves with more inducing points ($M$), and Cholesky and CIQ models have similar performance. However CIQ scales to larger values of $M$.

In Fig. S5 we plot the predictive error of CIQ-SVGP and Chol-SVGP models as a function of $M$. For the two regression datasets (3droad and Precipitation) error is measured by test set root mean squared error (RMSE). On the Covtype classification dataset error is measured by the test set $0/1$ loss. As with the NLL results in Fig. 3 we find that the CIQ-SVGP and Chol-SVGP perform similarly, despite the fact that CIQ-SVGP can be up to $5.6\times$ faster. Moreover, we see that error continuously decreases with more inducing points up to $M = 10,000$.

In Fig. S6 we plot the learned hyperparameters of the Precipitation SVGP models: 1) $o^2$ (the kernel outputscale)—which roughly corresponds to variance explained as "signal" in the data; 2) $\sigma^2_{\text{obs}}$—which roughly corresponds to variance explained away as observational noise; and 3) $\nu$ (degrees of freedom)—which controls the tails of the noise model (lower $\nu$ corresponds to heavier tails). As $M$ increases, we find that the observational noise parameter decreases by a factor of 4—down from 0.19 to 0.05—while the $\nu$ parameter also decreases. Models with larger $M$ values can more closely approximate the true posterior [39]; therefore, we expect that the parameters from the larger-$M$ likelihoods more closely correspond to the true dataset noise. This confirms findings from Bauer et al. [7], who argue that variational approximations with small $M$ can tend to overestimate the amount of noise in datasets.

Fig. S7 is a histogram displaying the msMINRES iterations needed to achieve a relative residual of $10^{-3}$ when training a $M = 5,000$ SVGP model on the 3droad dataset (subsampled to 30,000 data points). Most msMINRES calls converge in fewer than 100 iterations; almost no calls require more than 200 iterations. We hypothesize that this fast convergence is due to solving shifted systems

Figure S6: Hyperparameters versus number of inducing points ($M$) for Chol-SVGP and CIQ-SVGP (Precipitation dataset, Student-T likelihood). As $M$ increases, the kernel outputscale (left) also increases. At the same time, the estimated observational noise (middle) decreases as does the estimated degrees of freedom (right), reflecting a heavier-tailed noise distribution. This suggests that, with larger $M$, SVGP models can find more signal in the data.

Figure S7: Number of msMINRES iterations needed to achieve a relative residual of $10^{-3}$. Histogram captures training a $M = 5,000$ SVGP model on the 3droad dataset (subsampled to 30,000 data points).

$(\mathbf{K} + t_q\mathbf{I})$. The minimum eigenvalues of the shifted matrix are lower-bounded by $t_q$, and therefore shifted systems have a better condition number than the unshifted matrix $\mathbf{K}$.

## B   Quadrature for Matrix Square Roots

Here we briefly describe the quadrature formula derived by Hale et al. [35] for use with Cauchy's integral formula and refer the reader to the original publication for more details.

Assume that $\mathbf{K}$ is a positive definite matrix, and thus has real positive eigenvalues. Our goal is to approximate Cauchy's integral formula with a quadrature estimate:

$$f(\mathbf{K}) = \frac{1}{2\pi i} \oint_\Gamma f(\tau)\,(\tau\mathbf{I} - \mathbf{K})^{-1}\,\mathrm{d}\tau \tag{S1}$$

$$\approx \frac{1}{2\pi i} \sum_{q=1}^{Q} \widetilde{w}_q f(\tau_q)\,(\tau_q\mathbf{I} - \mathbf{K})^{-1}, \tag{S2}$$

where $f(\cdot)$ is analytic on and within $\Gamma$, and $\widetilde{w}_q$ and $\tau_q$ are quadrature weights and nodes respectively. Note that Eq. (S1) holds true for any closed contour $\Gamma$ in the complex plane that winds once (counterclockwise) around the spectrum of $\mathbf{K}$.

**A naïve approach with uniformly-spaced quadrature.** For now, assume that $\lambda_{\min}$ and $\lambda_{\max}$—the minimum and maximum eigenvalues of $\mathbf{K}$—are known. (We will later address how they can be efficiently estimated.) A naïve first approach to Eq. (S2) is to uniformly place the quadrature locations in a circle that surrounds the eigenvalues and avoids crossing the negative real axis, where we anticipate $f$ may be singular:

$$\tau_q = \frac{\lambda_{\max} + \lambda_{\min}}{2} + \frac{\lambda_{\max}}{2} e^{2i\pi(q/Q)}, \quad \widetilde{w}_q = \frac{1}{Q}, \quad q = 0, 1, \ldots, Q-1.$$

This corresponds to a standard trapezoid quadrature rule. However, Hale et al. [35] demonstrate that the convergence of this quadrature rule depends linearly on the condition number $\kappa(\mathbf{K}) = \lambda_{\max}/\lambda_{\min}$. In particular, this is because the integrand is only analytic in a narrow region around the chosen contour. As many kernel matrices tend to be approximately low-rank and therefore ill-conditioned, this simple quadrature rule requires large $Q$ to achieve the desired numerical accuracy.

**Improving convergence with conformal mappings.** Rather than uniformly spacing the quadrature points, it makes more sense to place more quadrature points near $\lambda_{\min}$ and fewer near $\lambda_{\max}$. This can be accomplished by using the above trapezoid quadrature rule in a transformed parameter space that is "stretched" near $\lambda_{\min}$ and contracted near $\lambda_{\max}$. Mathematically, this is accomplished by applying a conformal mapping that moves the singularities to the upper and lower boundaries of a periodic rectangle. We may then apply the trapezoid rule along a contour traversing the middle of the rectangle—maximizing the region in which the function we are integrating is analytic around the contour.

## B.1 A Specific Quadrature Formula for $f(\mathbf{K}) = \mathbf{K}^{-1/2}$

Hale et al. [35] suggest performing a change of variables that projects Eq. (S1) onto an annulus. Uniformly spaced quadrature points inside the annulus will cluster near $\lambda_{\min}$ when projected back into the complex plane. This change of variables has a simple analytic formula involving Jacobi elliptic functions (see [35, Sec. 2] for details.) In the special case of $f(\mathbf{K}) = \mathbf{K}^{-1/2}$, we can utilize an additional change of variables for an even more efficient quadrature formulation [35, Sec. 4]. Setting $\sigma = \tau^{1/2}$, we have

$$\mathbf{K}^{-\frac{1}{2}} = \frac{1}{\pi i} \oint_{\Gamma_s} \left(\sigma^2 \mathbf{I} - \mathbf{K}\right)^{-1} \mathrm{d}\sigma.$$

$$\approx \frac{1}{\pi i} \sum_{q=1}^{Q} \widetilde{w}_q \left(\sigma_q^2 \mathbf{I} - \mathbf{K}\right)^{-1}, \tag{S3}$$

where $\Gamma_\sigma$ is a contour that surrounds the spectrum of $\mathbf{K}^{1/2}$. Since the integrand is symmetric with respect to the real axis, we only need to consider the imaginary portion of $\Gamma_\sigma$. Consequently, all the $\tau_q$ quadrature locations (back in the original space) will be real-valued and negative. Combining this square-root change-of-variables with the annulus change-of-variables results in the following quadrature weights/locations:

$$\sigma_q^2 = \lambda_{\min} \left(\mathrm{sn}(iu_q \mathcal{K}'(k) \mid k)\right)^2,$$
$$\widetilde{w}_q = -\frac{2\sqrt{\lambda_{\min}}}{\pi Q} \left[\mathcal{K}'(k) \ \mathrm{cn}\left(iu_q \mathcal{K}'(k) \mid k\right) \ \mathrm{dn}\left(iu_q \mathcal{K}'(k) \mid k\right)\right], \tag{S4}$$

where we adopt the following notation:

- $k = \sqrt{\lambda_{\min}/\lambda_{\max}} = 1/\sqrt{\kappa(\mathbf{K})}$;
- $\mathcal{K}'(k)$ is the complete elliptic integral of the first kind with respect to the complimentary elliptic modulus $k' = \sqrt{1 - k^2}$;
- $u_q = \frac{1}{Q}(q - \frac{1}{2})$; and

- $\text{sn}(\cdot \mid k)$, $\text{cn}(\cdot \mid k)$, and $\text{dn}(\cdot \mid k)$ are the Jacobi elliptic functions with respect to elliptic modulus $k$.

The weights $\widetilde{w}_q$ and locations $\sigma_q^2$ from Eq. (S4) happen to be real-valued and negative. Setting $t_q = -\sigma_q^2$ and $w_q = -\widetilde{w}_q$ gives us:

$$\mathbf{K}^{-\frac{1}{2}} \approx \sum_{q=1}^{Q} w_q \left(t_q \mathbf{I} + \mathbf{K}\right)^{-1}, \quad w_q = -\widetilde{w}_q > 0, \quad t_q = -\sigma_q^2 > 0. \tag{S5}$$

An immediate consequence of this is that the shifted matrices $(t_q \mathbf{I} + \mathbf{K})$ are all positive definite.

**Convergence of the quadrature approximation.** Due to the double change-of-variables, the convergence of this quadrature rule in Eq. (S4) is extremely rapid—even for ill-conditioned matrices. Hale et al. prove the following error bound:

**Lemma 1** (Hale et al. [35], Thm. 4.1)**.** *Let $t_1, \ldots, t_Q > 0$ and $w_1, \ldots, w_Q > 0$ be the locations and weights of Hale et al.'s quadrature procedure. The error of Eq. (2) is bounded by:*

$$\left\| \mathbf{K} \sum_{q=1}^{Q} w_q \left(t_q \mathbf{I} + \mathbf{K}\right)^{-1} - \mathbf{K}^{\frac{1}{2}} \right\|_2 \leq \mathcal{O}\left( \exp\left( -\frac{2Q\pi^2}{\log \kappa(\mathbf{K}) + 3} \right) \right),$$

*where $\kappa(\mathbf{K}) = \lambda_{max}/\lambda_{min}$ is the condition number of $\mathbf{K}$.*

Remarkably, the error of Eq. (2) is *logarithmically* dependent on the conditioning of $\mathbf{K}$. Consequently, $Q \approx 8$ quadrature points is even sufficient for ill-conditioned matrices (e.g. $\kappa(\mathbf{K}) \approx 10^4$).

## B.2 Estimating the Minimum and Maximum Eigenvalues

The equations for the quadrature weights/locations depend on the extreme eigenvalues $\lambda_{\text{max}}$ and $\lambda_{\text{min}}$ of $\mathbf{K}$. Using the Lanczos algorithm [51]—which is a Krylov subspace method—we can obtain accurate estimates of these extreme eigenvalues using relatively few matrix-vector multiplies with $\mathbf{K}$.

**The Lanczos algorithm** is a method for computing an orthonormal basis for Krylov subspaces of a symmetric matrix $\mathbf{K}$ and, simultaneously, projections of $A$ onto that subspace. Given an initial vector $\mathbf{b}$, the algorithm iteratively factorizes $\mathbf{K}$ as:

$$\mathbf{K}\mathbf{Q}_J = \mathbf{Q}_J \mathbf{T}_J + \mathbf{r}_J \mathbf{e}_J^\top$$

where $\mathbf{e}_J$ is a unit vector, and

- $\mathbf{Q}_J \in \mathbb{R}^{N \times J}$ is an orthonormal basis of the $J^{\text{th}}$ Krylov subspace $\mathcal{K}(\mathbf{K}, \mathbf{b})$,
- $\mathbf{T}_J \in \mathbb{R}^{J \times J}$ is a symmetric tridiagonal matrix, and
- $\mathbf{r}_J \in \mathbb{R}^J$ is a residual term.

At a high level, the Lanczos iterations form the Krylov subspaces while simultaneously performing a process akin to modified Gram Schmidt orthogonalization:

$$\text{span}\{\mathbf{q}^{(1)}, \ldots, \mathbf{q}^{(J)}\} = \mathcal{K}(\mathbf{K}, \mathbf{b}) = \text{span}\{\mathbf{b}, \mathbf{K}\mathbf{b}, \mathbf{K}^2\mathbf{b}, \ldots, \mathbf{K}^{J-1}\mathbf{b}\}.$$

The orthogonal basis vectors are collected into $\mathbf{Q}$ and the orthogonalization coefficients are collected into $\mathbf{T}$. Due to the symmetry of $\mathbf{K}$ a three term recurrence exists for this process and each vector $\mathbf{q}^{(j)}$ only has to be orthogonalized against the two previous basis vectors $\mathbf{q}^{(j-1)}$, $\mathbf{q}^{(j-2)}$—resulting in a tridiagonal $\mathbf{T}$.

**Estimating Extreme Eigenvalues from Lanczos.** To estimate $\lambda_{\text{min}}$ and $\lambda_{\text{max}}$ from Lanczos, we perform an eigendecomposition of $\mathbf{T}_J$. If $J$ is small (i.e. $J \approx 10$) then this eigendecomposition requires minimal computational resources. In fact, as $\mathbf{T}_J$ is tridiagonal invoking standard routines allows computation of all the eigenvalues in $\mathcal{O}(J^2)$ time. A well-known convergence result of the Lanczos algorithm is that the extreme eigenvalues of $\mathbf{T}_J$ tend to converge rapidly to $\lambda_{\text{min}}$ and $\lambda_{\text{max}}$ [e.g. 31, 65]. Since the Lanczos algorithm always produces underestimates of the largest eigenavlue and overestimates of the smallest it is reasonable to use slightly larger and smaller values in the construction of the quadrature scheme—as we see in Lemma 1, the necessary number of quadrature nodes is insensitive to small overestimates of the condition number.

**Algorithm 2:** Computing $w_q$ and $t_q$ for Contour Integral Quadrature

**Input** : `mvm_`$\mathbf{K}$ `(·)` – function for matrix-vector multiplication (MVM) with matrix $\mathbf{K}$
$Q$ – number of quad. points

**Output:** $w_1, \ldots, w_Q, t_1, \ldots, t_Q$

```
// Estimate extreme eigenvalues with Lanczos.
```
$\_, \mathbf{T} \leftarrow$ `lanczos(mvm_`$\mathbf{K}$`(·))` `// Lanczos w/ rand.  init.  vector`
$\lambda_{\min}, \cdots, \lambda_{\max} \leftarrow$ `symeig(`$\mathbf{T}$`)`

```
// Compute elliptic integral of the first kind.
// We use the relation
```
$\mathcal{K}'(k) = \mathcal{K}(k')$`, where` $k' = \sqrt{1 - k^2}$ `is the`
   `complementary elliptic modulus.`
$k^2 \leftarrow \lambda_{\min}/\lambda_{\max}$ `// The squared elliptic modulus.`
$k'^2 \leftarrow \sqrt{1 - k^2}$ `// The squared complementary elliptic modulus.`
`K'` $\leftarrow$ `ellipke(`$k'^2$`)` `// K' =` $\mathcal{K}'(k)$

```
// Compute each quadrature weight/location.
```
**for** $q \leftarrow 1$ **to** $Q$ **do**
    $u_q \leftarrow (q - 1/2)/Q$
    `// Compute Jacobi elliptic fn's via Jacobi's imaginary transform.`
    `// First we compute` $\overline{\mathrm{sn}}_q = \mathrm{sn}(u_q\mathcal{K}'(k)|k')$`,` $\overline{\mathrm{cn}}_q = \mathrm{cn}(u_q\mathcal{K}'(k)|k')$`,`
       $\overline{\mathrm{dn}}_q = \mathrm{dn}(u_q\mathcal{K}'(k)|k')$`.`
    $\overline{\mathrm{sn}}_q, \overline{\mathrm{cn}}_q, \overline{\mathrm{dn}}_q \leftarrow$ `ellipj(`$u_q$`K'`$, k'^2$`)`
    `// Use identities to convert` $\overline{\mathrm{sn}}_q$`,` $\overline{\mathrm{cn}}_q$`,` $\overline{\mathrm{dn}}_q$ `values into`
    `//` $\mathrm{sn}_q = \mathrm{sn}(iu_q\mathcal{K}'(k)|k)$`,` $\mathrm{cn}_q = \mathrm{cn}(iu_q\mathcal{K}'(k)|k)$`,` $\mathrm{dn}_q = \mathrm{dn}(iu_q\mathcal{K}'(k)|k)$`.`
    $\mathrm{sn}_q \leftarrow i\left[\overline{\mathrm{sn}}_q/\overline{\mathrm{cn}}_q\right]$
    $\mathrm{dn}_q \leftarrow \left[\overline{\mathrm{dn}}_q/\overline{\mathrm{cn}}_q\right]$
    $\mathrm{cn}_q \leftarrow \left[1/\overline{\mathrm{cn}}_q\right]$
    `// Quadrature weight` $w_q$ `and location` $t_q$
    $w_q \leftarrow (-2\lambda_{\min}^{1/2})/(\pi Q)\,\mathrm{K}'\,\mathrm{cn}_q\,\mathrm{dn}_q$
    $t_q \leftarrow \lambda_{\min}\left(\mathrm{sn}_q\right)^2$
**end**
**return** $w_1, \ldots, w_Q, t_1, \ldots, t_Q$

## B.3 The Complete Quadrature Algorithm

Alg. 2 obtains the quadrature weights $w_q$ and locations $t_q$ corresponding to Eqs. (S4) and (S5). Computing these weights requires $\approx 10$ matrix-vector multiplies with $\mathbf{K}$—corresponding to the Lanczos iterations—for a total time complexity of $\mathcal{O}(N)$. All computations involving elliptic integrals can be readily computed using routines available in e.g. the SciPy library.

## C  The msMINRES Algorithm

Before introducing the msMINRES algorithm, we will first introduce MINRES as proposed by Paige and Saunders [59]; MINRES can be derived from the Lanczos algorithm [51] and, therefore, is able to take advantage of the same three term vector recurrence when building the necessary Krylov subspaces. We will then describe how msMINRES can be derived as a straightforward extension. Notably, we present this section assuming our best initial guess for the linear system we seek to solve is zero. If this is not the case a single step of iterative refinement can be used and the resulting residual system is solved with zero as the initial guess.

### C.1  Standard MINRES

The method of minimum residuals (MINRES) [59] is an alternative to linear conjugate gradients, with the advantage that it can be applied to indefinite and singular symmetric matrices $\mathbf{K}$. Paige and Saunders [59] formulate MINRES to solve the least-squares problem $\arg\min_{\mathbf{c}} \|\mathbf{K}\mathbf{c} - \mathbf{b}\|_2$. Each

iteration $J$ produces a solution $\mathbf{c}_J$ which is optimal within the $J^{\text{th}}$ Krylov subspace:

$$\mathbf{c}_J^{(\text{MINRES})} = \underset{\mathbf{c}\in\mathcal{K}_J(\mathbf{K},\mathbf{b})}{\arg\min} \|\mathbf{K}\mathbf{c} - \mathbf{b}\|_2. \tag{S6}$$

Using the Lanczos matrices and some mathematical manipulation, Eq. (S6) can be re-formulated as an unconstrained optimization problem:

$$\mathbf{c}_J^{(\text{MINRES})} = \|\mathbf{b}\|_2 \mathbf{Q}_J \mathbf{z}_J$$

$$\mathbf{z}_J = \underset{\mathbf{y}\in\mathbb{R}^J}{\arg\min} \left\|\left(\widetilde{\mathbf{T}}_J\right)\mathbf{y} - \mathbf{e}_1\right\|_2, \quad \widetilde{\mathbf{T}}_J = \begin{bmatrix} \mathbf{T}_J \\ \|\mathbf{r}_J\|_2 \mathbf{e}_J^\top \end{bmatrix}, \tag{S7}$$

where $\mathbf{e}_1, \mathbf{e}_J$ are unit vectors, and $\mathbf{Q}_J$, $\mathbf{T}_J$, and $\mathbf{r}_J$ are the outputs from the Lanczos algorithm. Since Eq. (S7) is a least-squares problem (guaranteed to be full column-rank unless $\mathbf{b}$ lives in the $J^{\text{th}}$ Krylov subspace—at which point we would exactly solve the problem), we can write the analytic solution to it using the reduced QR factorization of $\widetilde{\mathbf{T}}_J = \mathcal{Q}_J \mathbf{R}_J$ [e.g. 31]:

$$\mathbf{c}_J^{(\text{MINRES})} = \|\mathbf{b}\|_2 \, \mathbf{Q}_J \left(\mathbf{R}^{-1}\mathcal{Q}_J^\top\right) \mathbf{e}_1. \tag{S8}$$

One way to perform MINRES is first running $J$ iterations of the Lanczos algorithm, computing $\widetilde{\mathbf{T}}_J = \mathcal{Q}_J \mathbf{R}_J$, and then plugging the resulting $\mathbf{Q}_J$, $\mathcal{Q}_J$, and $\mathbf{R}_J$ into Eq. (S8). However, this is unsatisfactory as, naïvely it requires storing the $N \times J$ matrix $\mathbf{Q}_J$ [e.g. 31] so that $\mathbf{c}_J$ can be formed. Paige and Saunders instead introduce a vector recurrence to iteratively compute $\mathbf{c}_J^{(\text{MINRES})}$. This is possible because the QR factorizations of of successive $\widetilde{\mathbf{T}}_J$ may be related, allowing for the derivation of a simple update $\mathbf{c}_{J-1} \to \mathbf{c}_J$. This recurrence relation, which is given by Alg. 3 and broadly described below is exactly equivalent to Eq. (S8); however it uses careful bookkeeping to avoid storing any $N \times J$ terms.

First we note that the $\widetilde{\mathbf{T}}_J$ matrices are formed recursively, and thus their QR factorizations are also recursive:

$$\mathcal{Q}^\top \widetilde{\mathbf{T}}_J = \begin{bmatrix} \mathcal{Q}_{J-1}^\top & \mathcal{Q}^{\top(J,1:J-1)} \\ \mathcal{Q}^{\top(1:J-1,J+1)} & \mathcal{Q}^{(J,J+1)} \end{bmatrix} \begin{bmatrix} \widetilde{\mathbf{T}}_{J-1} & \mathbf{t}^{(J)} \\ \mathbf{0}^\top & \|\mathbf{r}_J\| \end{bmatrix} = \begin{bmatrix} \mathbf{R}_{J-1} & \mathbf{r}^{(J,1:J-1)} \\ \mathbf{0} & R^{(J,J)} \end{bmatrix} = \mathbf{R}_J$$

where $\mathbf{t}^{(J)}$ and $[\mathbf{r}^{(J,1:J-1)}; R^{(J,J)}]$ are the last columns of $\mathbf{T}_J$ and $\mathbf{R}_J$ respectively. Moreover, if we recursively form $\mathbf{R}_J^{-1}$ as

$$\mathbf{R}_J^{-1} = \begin{bmatrix} \mathbf{R}_{J-1} & \mathbf{r}^{(J,1:J-1)} \\ \mathbf{0} & R^{(J,J)} \end{bmatrix}^{-1} = \begin{bmatrix} \mathbf{R}_{J-1}^{-1} & \left(\mathbf{R}_{J-1}^{-1}\mathbf{r}^{(J,1:J-1)}\right)/R^{(J,J)} \\ \mathbf{0} & 1/R^{(J,J)} \end{bmatrix},$$

then Eq. (S8) can be re-written in a decent-style update:

$$\mathbf{c}_J^{(\text{MINRES})} = \|\mathbf{b}\|_2 \begin{bmatrix} \mathbf{Q}_{J-1} & \mathbf{q}^{(J)} \end{bmatrix} \begin{bmatrix} \mathbf{R}_{J-1}^{-1} & \frac{\mathbf{R}_{J-1}^{-1}\mathbf{r}^{(J,1:J-1)}}{R^{(J,J)}} \\ \mathbf{0} & 1/R^{(J,J)} \end{bmatrix} \begin{bmatrix} \mathcal{Q}_{J-1}^\top & \mathcal{Q}^{\top(J,1:J-1)} \\ \mathcal{Q}^{\top(1:J-1,J+1)} & \mathcal{Q}^{(J,J+1)} \end{bmatrix} \mathbf{e}_1$$

$$= \|\mathbf{b}\|_2 \begin{bmatrix} \mathbf{Q}_{J-1}\mathbf{R}_{J-1}^{-1} & \frac{\mathbf{Q}_{J-1}\mathbf{R}_{J-1}^{-1}\mathbf{r}^{(J,1:J-1)}}{R^{(J,J)}} \\ \mathbf{0} & 1/R^{(J,J)}\mathbf{q}_{J-1} \end{bmatrix} \begin{bmatrix} \mathcal{Q}_{J-1}^\top \mathbf{e}_1 \\ \mathcal{Q}^{\top(1,J+1)} \end{bmatrix}$$

$$= \underbrace{\left(\|\mathbf{b}\|_2 \mathbf{Q}_{J-1}\mathbf{R}_{J-1}^{-1}\mathcal{Q}_{J-1}\mathbf{e}_1\right)}_{\mathbf{c}_{J-1}^{(\text{MINRES})}} + \underbrace{\frac{\|\mathbf{b}\|_2 \mathcal{Q}^{\top(1,J+1)}}{R^{(J,J)}}}_{\varphi_J} \underbrace{\begin{bmatrix} \mathbf{Q}_{J-1}\mathbf{R}_{J-1}^{-1}\mathbf{r}^{(J,1:J-1)} \\ \mathbf{q}_{J-1} \end{bmatrix}}_{\mathbf{d}_J}. \tag{S9}$$

Thus $\mathbf{c}_J^{(\text{MINRES})} = \mathbf{c}_{J-1}^{(\text{MINRES})} + \varphi_J \mathbf{d}_J$. The only seemingly expensive part of this update is computing $\mathbf{d}_J$, as we need to compute $\mathbf{Q}_{J-1}\mathbf{R}_{J-1}^{-1}\mathbf{r}^{(J,1:J-1)}$. $\mathbf{r}^{(J,1:J-1)}$, which is the next entry in the QR factorization of $\widetilde{\mathbf{T}}_J$, can be cheaply computed using Givens rotations (see [e.g. 31, Ch. 11.4.1]). Moreover, only the last two entries of $\mathbf{r}^{(J,1:J-1)}$ will be non-zero (due to the tridiagonal structure of $\widetilde{\mathbf{T}}_J$). Consequently, we only need to store the last two vectors of $\mathbf{Q}_{J-1}\mathbf{R}_{J-1}^{-1}$, which again can be computed recursively.

In total, the whole procedure only requires the storage of $\approx 6$ vectors. Each iteration requires a single MVM with $\mathbf{K}$ (to form the next Lanczos vector $\mathbf{q}_J$); and all subsequent operations are $\mathcal{O}(N)$. The entire procedure is given by Alg. 3. For simplicity, we have presented the algorithm as if run for a fixed number of steps $J$. In practice, the MINRES procedure admits inexpensive computation of the residual at each iteration [59] allowing for robust stopping criteria to be used.

**Algorithm 3:** Method of Minimum Residuals (MINRES).

**Input** : $\texttt{mvm\_K}(\cdot)$ – function for MVM with matrix $\mathbf{K}$
&emsp;&emsp;&emsp; $\mathbf{b}$ – vector to solve against
**Output** : $\mathbf{c} = \mathbf{K}^{-1}\mathbf{b}$.

$\mathbf{c}_1 \leftarrow \mathbf{0}$ // Current solution.
$\mathbf{d}_1, \mathbf{d}_0 \leftarrow \mathbf{0}$ // Current & prev. "search" direction.
$\varphi_2 \leftarrow \|\mathbf{b}\|_2$ // Current "step" size.

$\mathbf{q}_1 \leftarrow \mathbf{b}/\|\mathbf{b}\|_2$ // Current Lanczos vector.
$\mathbf{v}_1 \leftarrow \texttt{mvm\_K}(\mathbf{q}_0)$ // Buffer for MVM output.
$\delta_1 \leftarrow \|\mathbf{b}\|_2$ // Current Lanczos residual/sub-diagonal.
$\delta_0 \leftarrow 1$ // Prev. Lanczos residual/sub-diagonal.
$\eta_1 \leftarrow 1$ // Current scaling term.
$\eta_0 \leftarrow 0$ // Prev. scaling term.

**for** $j \leftarrow 2$ **to** $J$ **do**
&emsp;// Run one iter of Lanczos. Gets next vector of $\mathbf{Q}$ matrix, and next
&emsp;&emsp; diag/sub-diag $(\gamma, \delta)$ entries of $\mathbf{T}$ matrix.
&emsp;$\mathbf{q}_j \leftarrow \mathbf{v}_j/\delta_j$
&emsp;$\mathbf{v}_j \leftarrow \texttt{mvm\_K}(\mathbf{q}_j) - \delta_j\mathbf{q}_{j-1}$
&emsp;$\gamma_j \leftarrow \mathbf{q}_j\mathbf{v}_j$
&emsp;$\mathbf{v}_j \leftarrow \mathbf{v}_j - \gamma_j\mathbf{q}_j$
&emsp;$\delta_j \leftarrow \|\mathbf{v}_j\|$
&emsp;// Compute the next $\mathbf{r}^{(J)}$ (part of QR) via Givens rotations. There
&emsp;&emsp; are three non-0 entries: $\mathbf{R}^{(J,J-2:J)} = [\epsilon_J, \zeta_J, \eta_J]$.
&emsp;$\epsilon_j \leftarrow \delta_{j-1}\left(\delta_{j-2}/\sqrt{\delta_{j-2}^2 + \eta_{j-2}^2}\right)$
&emsp;$\zeta_j \leftarrow \delta_{j-1}\left(\eta_{j-2}/\sqrt{\delta_{j-2}^2 + \eta_{j-2}^2}\right)$
&emsp;$\eta_j \leftarrow \gamma_j\left(\eta_{j-1}/\sqrt{\delta_{j-1}^2 + \eta_{j-1}^2}\right) + \zeta_j\left(\delta_{j-1}/\sqrt{\delta_{j-1}^2 + \eta_{j-1}^2}\right)$
&emsp;$\zeta_j \leftarrow \zeta_j\left(\eta_{j-1}/\sqrt{\delta_{j-1}^2 + \eta_{j-1}^2}\right) + \gamma_j\left(\delta_{j-1}/\sqrt{\delta_{j-1}^2 + \eta_{j-1}^2}\right)$
&emsp;$\eta_j \leftarrow \eta_j\left(\eta_j/\sqrt{\delta_j^2 + \eta_j^2}\right)$
&emsp;// Compute "step" size $\varphi_J = \mathcal{Q}^{(1,J+1)}/R^{(J,J)}$.
&emsp;$\varphi_j \leftarrow \varphi_{j-1}\left(\delta_{j-1}/\sqrt{\delta_{j-1}^2 + \eta_{j-1}^2}\right)\left(\eta_j/\sqrt{\delta_j^2 + \eta_j^2}\right)$
&emsp;// Update the current solution based on the $\mathbf{r}^{(J)}$ entries $(\epsilon_J, \zeta_J, \eta_J)$
&emsp;&emsp; and previous search vectors $\mathbf{d}_{j-1}$, $\mathbf{d}_{j-2}$.
&emsp;$\mathbf{d}_j \leftarrow (\mathbf{q} - \zeta_j\mathbf{d}_{j-1} - \epsilon_j\mathbf{d}_{j-2})/\eta_j$
&emsp;$\mathbf{c}_j \leftarrow \mathbf{c}_{j-1} + \varphi_j\mathbf{d}_j$
**end**

**return** $\|\mathbf{b}\|_2\,\mathbf{c}_j$

---

### C.2 Multi-Shift MINRES (msMINRES)

To adapt MINRES to multiple shifts (i.e. msMINRES), we exploit a well-established fact about the shift invariance of Krylov subspaces (see [e.g. 17, 24, 46, 65]).

**Observation 1.** *Let* $\mathbf{K}\mathbf{Q}_J = \mathbf{Q}_J\mathbf{T}_J + \mathbf{r}_J\mathbf{e}_J^\top$ *be the Lanczos factorization for* $\mathbf{K}$ *given the initial vector* $\mathbf{b}$. *Then*

$$(\mathbf{K} + t\mathbf{I})\mathbf{Q}_J = \mathbf{Q}_J(\mathbf{T}_J + t\mathbf{I}) + \mathbf{r}_J\mathbf{e}_J^\top$$

*is the Lanczos factorization for matrix* $(\mathbf{K} + t\mathbf{I})$ *with initial vector* $\mathbf{b}$.

In other words, if we run Lanczos on $\mathbf{K}$ and $\mathbf{b}$, then we get the Lanczos factorization of $(\mathbf{K} + t\mathbf{I})$ *for free*, without any additional MVMs! Consequently, we can re-use the $\mathbf{Q}_J$ and $\mathbf{T}_J$ Lanczos matrices

**Algorithm 4:** Multi-shift MINRES (msMINRES). Differences from MINRES (Alg. 3) are in blue. Blue `for` loops are parallelizable.

**Input** : `mvm_K`($\cdot$) – function for MVM with matrix $\mathbf{K}$
$\qquad$ $\mathbf{b}$ – vector to solve against
$\qquad$ $t_1, \ldots, t_Q$ – shifts
**Output** : $\mathbf{c}_1 = (\mathbf{K} + t_1)^{-1}\mathbf{b}, \ldots, \mathbf{c}_Q = (\mathbf{K} + t_Q)^{-1}\mathbf{b}$.

$\mathbf{q}_1 \leftarrow \mathbf{b}/\|\mathbf{b}\|_2$ // Current Lanczos vector.
$\mathbf{v}_1 \leftarrow$ `mvm_K`( $\mathbf{q}_0$) // Buffer for MVM output.
$\delta_1 \leftarrow \|\mathbf{b}\|_2, \delta_0 \leftarrow 1$ // Current/prev. Lanczos residual/sub-diagonal.
**for** $q \leftarrow 1$ **to** $Q$ **do**
$\quad$ $\mathbf{c}_1^{(q)} \leftarrow \mathbf{0}$ // Current solution.
$\quad$ $\mathbf{d}_1^{(q)}, \mathbf{d}_0^{(q)} \leftarrow \mathbf{0}$ // Current & prev. "search" direction.
$\quad$ $\varphi_2^{(q)} \leftarrow \|\mathbf{b}\|_2$ // Current "step" size.
$\quad$ $\eta_1^{(q)} \leftarrow 1, \eta_0^{(q)} \leftarrow 0$ // Current/prev. scaling term.
**end**
**for** $j \leftarrow 2$ **to** $J$ **do**
$\quad$ $\mathbf{q}_j \leftarrow \mathbf{v}_j/\delta_j$
$\quad$ $\mathbf{v}_j \leftarrow$ `mvm_K`( $\mathbf{q}_j$) $-\delta_j\mathbf{q}_{j-1}$
$\quad$ $\gamma_j \leftarrow \mathbf{q}_j\mathbf{v}_j$
$\quad$ $\mathbf{v}_j \leftarrow \mathbf{v}_j - \gamma_j\mathbf{q}_j$
$\quad$ $\delta_j \leftarrow \|\mathbf{v}_j\|$
$\quad$ **for** $q \leftarrow 1$ **to** $Q$ **do**
$\qquad$ $\epsilon_j^{(q)} \leftarrow \delta_{j-1}\left(\delta_{j-2}/\sqrt{\delta_{j-2}^2 + \eta_{j-2}^{(q)2}}\right)$
$\qquad$ $\zeta_j^{(q)} \leftarrow \delta_{j-1}\left(\eta_{j-2}^{(q)}/\sqrt{\delta_{j-2}^2 + \eta_{j-2}^{(q)2}}\right)$
$\qquad$ $\eta_j^{(q)} \leftarrow (\gamma_j + t_q)\left(\eta_{j-1}^{(q)}/\sqrt{\delta_{j-1}^2 + \eta_{j-1}^{(q)2}}\right) + \zeta_j^{(q)}\left(\delta_{j-1}/\sqrt{\delta_{j-1}^2 + \eta_{j-1}^{(q)2}}\right)$
$\qquad$ $\zeta_j^{(q)} \leftarrow \zeta_j^{(q)}\left(\eta_{j-1}^{(q)}/\sqrt{\delta_{j-1}^2 + \eta_{j-1}^{(q)2}}\right) + (\gamma_j + t_q)\left(\delta_{j-1}/\sqrt{\delta_{j-1}^2 + \eta_{j-1}^{(q)2}}\right)$
$\qquad$ $\eta_j^{(q)} \leftarrow \eta_j^{(q)}\left(\eta_j^{(q)}/\sqrt{\delta_j^2 + \eta_j^{(q)2}}\right)$
$\qquad$ $\varphi_j^{(q)} \leftarrow \varphi_{j-1}^{(q)}\left(\delta_{j-1}/\sqrt{\delta_{j-1}^2 + \eta_{j-1}^{(q)2}}\right)\left(\eta_j^{(q)}/\sqrt{\delta_j^2 + \eta_j^{(q)2}}\right)$
$\qquad$ $\mathbf{d}_j^{(q)} \leftarrow \left(\mathbf{q} - \zeta_j^{(q)}\mathbf{d}_{j-1}^{(q)} - \epsilon_j^{(q)}\mathbf{d}_{j-2}^{(q)}\right)/\eta_j^{(q)}$
$\qquad$ $\mathbf{c}_j^{(q)} \leftarrow \mathbf{c}_{j-1}^{(q)} + \varphi_j^{(q)}\mathbf{d}_j^{(q)}$
$\quad$ **end**
**end**
**return** $\|\mathbf{b}\|_2\, \mathbf{c}_j$

to compute *multiple shifted solves*.

$$(\mathbf{K} + t\mathbf{I})^{-1}\mathbf{b} \approx \|\mathbf{b}\|_2\, \mathbf{Q}_J\left(\mathbf{R}_J^{(t)-1}\boldsymbol{\mathcal{Q}}_J^{(t)\top}\right)\mathbf{e}_1, \quad \boldsymbol{\mathcal{Q}}_J^{(t)}\mathbf{R}_J^{(t)} = \begin{bmatrix} \mathbf{T}_J + t\mathbf{I} \\ \|\mathbf{r}_J\|_2\mathbf{e}_J^\top \end{bmatrix}, \qquad \text{(S10)}$$

Assuming $\mathbf{Q}$ and $\mathbf{T}$ have been previously computed, Eq. (S10) requires no additional MVMs with $\mathbf{K}$. We refer to this multi-shift formulation as **Multi-Shift MINRES**, or **msMINRES**.

**A simple vector recurrence for msMINRES.** Just as with standard MINRES, Eq. (S10) can also be computed via a vector recurrence. We can derive a msMINRES algorithm simply by modifying the existing MINRES recurrence. Before the QR step in Alg. 3, we add $t$ to the Lanczos diagonal terms ($\gamma_j + t$, where $\gamma_j = T^{(j,j)}$). This can be extended to simultaneously handle *multiple shifts* $t_1, \ldots, t_Q$. Each shift would compute its own QR factorization, its own step size $\varphi_j^{(t_q)}$, and its own

search vector $\mathbf{d}_j^{(t_q)}$. However, all shifts share the same Lanczos vectors $\mathbf{q}_j$ and therefore share the same MVMs. The operations for each shift can be vectorized for efficient parallelization.

To summarize: the resulting algorithm—msMINRES—gives us approximations to $(t_1\mathbf{I} + \mathbf{K})^{-1}\mathbf{b}$, $\ldots$, $(t_Q\mathbf{I} + \mathbf{K})^{-1}$ *essentially for free* by leveraging the information we needed anyway to compute $\mathbf{K}^{-1}\mathbf{b}$. Alg. 4 outlines the procedure; below we re-highlight its computational properties:

**Property 1 (Restated)** (Computation/Memory of msMINRES-CIQ). *$J$ iterations of msMINRES requires exactly $J$ matrix-vector multiplications (MVMs) with the input matrix $\mathbf{K}$, regardless of the number of quadrature points $Q$. The resulting runtime of msMINRES-CIQ is $\mathcal{O}(J\xi(\mathbf{K}))$, where $\xi(\mathbf{K})$ is the time to perform an MVM with $\mathbf{K}$. The memory requirement is $\mathcal{O}(QN)$ in addition to what's required to store $\mathbf{K}$.*

## D   Preconditioning msMINRES-CIQ

To improve the convergence of Thm. 1, we can introduce a preconditioner $\mathbf{P}$ where $\mathbf{P}^{-1}\mathbf{K} \approx \mathbf{I}$. For standard MINRES, applying a preconditioner is straightforward. We simply use MINRES to solve the system

$$\left(\mathbf{P}^{-1/2}\mathbf{K}\mathbf{P}^{-1/2}\right)\mathbf{P}^{1/2}\mathbf{c} = \mathbf{P}^{-1/2}\mathbf{b},$$

which has the same solution $\mathbf{c}$ as the original system. In practice the preconditioned MINRES vector recurrence does not need access to $\mathbf{P}^{-1/2}$—it only needs access to $\mathbf{P}^{-1}$ (see [12, Ch. 3.4] for details).

However, it is not immediately straightforward to apply preconditioning to msMINRES, as preconditioners break the shift-invariance property that is necessary for the $\mathcal{O}(JN^2)$ shifted solves [3, 46]. More specifically, if we apply $\mathbf{P}$ to msMINRES, then we obtain the solves

$$\mathbf{P}^{-1/2}(\mathbf{P}^{-1/2}\mathbf{K}\mathbf{P}^{-1/2} + t_q\mathbf{I})^{-1}(\mathbf{P}^{-1/2}\mathbf{b}).$$

Plugging these shifted solves into the quadrature equation Eq. (2) therefore gives us

$$\widetilde{\mathbf{a}}_J \approx \mathbf{P}^{-\frac{1}{2}}(\mathbf{P}^{-\frac{1}{2}}\mathbf{K}\mathbf{P}^{-\frac{1}{2}})^{-\frac{1}{2}}(\mathbf{P}^{-\frac{1}{2}}\mathbf{b}). \tag{S11}$$

In general, we cannot recover $\mathbf{K}^{-1/2}$ from Eq. (S11). Nevertheless, we can still obtain preconditioned solutions that are equivalent to $\mathbf{K}^{-1/2}\mathbf{b}$ and $\mathbf{K}^{1/2}\mathbf{b}$ up to an orthogonal rotation. Let $\mathbf{R} = \mathbf{K}\mathbf{P}^{-1/2}(\mathbf{P}^{-1/2}\mathbf{K}\mathbf{P}^{-1/2})^{-1/2}$. We have that

$$\mathbf{R}\mathbf{R}^\top = \mathbf{K}\left(\mathbf{P}^{-\frac{1}{2}}(\mathbf{P}^{-\frac{1}{2}}\mathbf{K}\mathbf{P}^{-\frac{1}{2}})^{-\frac{1}{2}}\right)\left((\mathbf{P}^{-\frac{1}{2}}\mathbf{K}\mathbf{P}^{-\frac{1}{2}})^{-\frac{1}{2}}\mathbf{P}^{-\frac{1}{2}}\right)\mathbf{K} = \mathbf{K}.$$

Thus $\mathbf{R}$ is equivalent to $\mathbf{K}^{1/2}$ up to orthogonal rotation. We can compute $\mathbf{R}\mathbf{b}$ (e.g. for sampling) by applying Eq. (S11) to the initial vector $\mathbf{P}^{1/2}\mathbf{b}$:

$$\mathbf{R}\mathbf{b} = \mathbf{K}\underbrace{\left[\mathbf{P}^{-\frac{1}{2}}(\mathbf{P}^{-\frac{1}{2}}\mathbf{K}\mathbf{P}^{-\frac{1}{2}})^{-\frac{1}{2}}\mathbf{P}^{-\frac{1}{2}}\right]\left(\mathbf{P}^{\frac{1}{2}}\mathbf{b}\right)}_{\text{Applying preconditioned msMINRES to } \mathbf{P}^{1/2}\mathbf{b}}. \tag{S12}$$

Similarly, $\mathbf{R}' = \mathbf{P}^{-1/2}\left(\mathbf{P}^{-1/2}\mathbf{K}\mathbf{P}^{-1/2}\right)^{-1/2}$ is equivalent to $\mathbf{K}^{-1/2}$ up to orthogonal rotation:

$$\mathbf{R}'\mathbf{R}'^\top = \left(\mathbf{P}^{-\frac{1}{2}}(\mathbf{P}^{-\frac{1}{2}}\mathbf{K}\mathbf{P}^{-\frac{1}{2}})^{-\frac{1}{2}}\right)\left((\mathbf{P}^{-\frac{1}{2}}\mathbf{K}\mathbf{P}^{-\frac{1}{2}})^{-\frac{1}{2}}\mathbf{P}^{-\frac{1}{2}}\right) = \mathbf{K}^{-1}.$$

We can compute $\mathbf{R}'\mathbf{b}$ (e.g. for whitening) via:

$$\mathbf{R}'\mathbf{b} = \underbrace{\left[\mathbf{P}^{-\frac{1}{2}}(\mathbf{P}^{-\frac{1}{2}}\mathbf{K}\mathbf{P}^{-\frac{1}{2}})^{-\frac{1}{2}}\mathbf{P}^{-\frac{1}{2}}\right]\left(\mathbf{P}^{\frac{1}{2}}\mathbf{b}\right)}_{\text{Applying preconditioned msMINRES to } \mathbf{P}^{1/2}\mathbf{b}}. \tag{S13}$$

Crucially, the convergence of Eqs. (S12) and (S13) depends on the conditioning $\kappa(\mathbf{P}^{-1}\mathbf{K}) \ll \kappa(\mathbf{K})$.

As with standard MINRES, msMINRES only requires access to $\mathbf{P}^{-1}$, not $\mathbf{P}^{-1/2}$. Note however that Eqs. (S12) and (S13) both require multiplies with $\mathbf{P}^{1/2}$. If a preconditioner $\mathbf{P}$ does not readily decompose into $\mathbf{P}^{1/2}\mathbf{P}^{1/2}$, we can simply run the CIQ algorithm on $\mathbf{P}$ to compute $\mathbf{P}^{1/2}\mathbf{b}$. Thus our requirements for a preconditioner are:

1) it affords efficient solves (ideally $o(N^2)$), and

2) it affords efficient MVMs (also ideally $o(N^2)$) for computing $\mathbf{P}^{1/2}\mathbf{b}$ via CIQ.

In our experiments we use the partial pivoted Cholesky preconditioner proposed by Gardner et al. [29], which satisfies the above requirements. The form of $\mathbf{P}$ is $\bar{\mathbf{L}}\bar{\mathbf{L}}^\top + \sigma^2\mathbf{I}$, where $\bar{\mathbf{L}}$ is a low-rank factor (produced by the partial pivoted Cholesky factorization [37]) and $\sigma^2\mathbf{I}$ is a small diagonal component. This preconditioner affords $\approx \mathcal{O}(N)$ MVMs by exploiting its low rank structure and $\approx \mathcal{O}(N)$ solves using the matrix inversion lemma. Moreover, this preconditioner is highly effective on many Gaussian covariance matrices [29, 79].

# E $\quad \mathcal{O}(M^2)$ Natural Gradient Updates

When performing variational inference, we must optimize the $\mathbf{m}'$ and $\mathbf{S}'$ parameters of the whitened variational distribution $q(\mathbf{u}') = \mathcal{N}(\mathbf{m}', \mathbf{S}')$. Rather than using standard gradient descent methods on these parameters, many have suggested that **natural gradient descent (NGD)** is better suited for variational inference [38, 45, 66]. NGD performs the following update:

$$\begin{bmatrix} \mathbf{m}' & \mathbf{S}' \end{bmatrix} \leftarrow \begin{bmatrix} \mathbf{m}' & \mathbf{S}' \end{bmatrix} - \varphi \boldsymbol{\mathcal{F}}^{-1} \begin{bmatrix} \frac{\partial \text{ELBO}}{\partial \mathbf{m}'} & \frac{\partial \text{ELBO}}{\partial \mathbf{S}'} \end{bmatrix} \tag{S14}$$

where $\varphi$ is a step size, $\begin{bmatrix} \frac{\partial \text{ELBO}}{\partial \mathbf{m}'} & \frac{\partial \text{ELBO}}{\partial \mathbf{S}'} \end{bmatrix}$ is the ELBO gradient, and $\boldsymbol{\mathcal{F}}$ is the *Fisher information matrix* of the variational parameters. Conditioning the gradient with $\boldsymbol{\mathcal{F}}^{-1}$ results in descent directions that are better suited towards distributional parameters [45].

For Gaussian distributions (and other exponential family distributions) the Fisher information matrix does not need to be explicitly computed. Instead, there is a simple closed-form update that relies on different parameterizations of the Gaussian $\mathcal{N}(\mathbf{m}', \mathbf{S}')$:

$$\begin{bmatrix} \boldsymbol{\theta} & \boldsymbol{\Theta} \end{bmatrix} \leftarrow \begin{bmatrix} \boldsymbol{\theta} & \boldsymbol{\Theta} \end{bmatrix} - \varphi \begin{bmatrix} \frac{\partial \text{ELBO}}{\partial \boldsymbol{\eta}} & \frac{\partial \text{ELBO}}{\partial \mathbf{H}} \end{bmatrix}. \tag{S15}$$

$[\boldsymbol{\theta}, \ \boldsymbol{\Theta}]$ are the Gaussian's *natural parameters* and $[\boldsymbol{\eta}, \ \mathbf{H}]$ are the Gaussian's *expectation parameters*:

$$\boldsymbol{\theta} = \mathbf{S}'^{-1}\mathbf{m}', \quad \boldsymbol{\Theta} = -\frac{1}{2}\mathbf{S}'^{-1},$$

$$\boldsymbol{\eta} = \mathbf{m}', \quad \mathbf{H} = \mathbf{m}'\mathbf{m}'^\top + \mathbf{S}'$$

In many NGD implementations, it is common to store the variational parameters via their natural representation $(\boldsymbol{\theta}, \boldsymbol{\Theta})$, compute the ELBO via the standard parameters $(\mathbf{m}', \mathbf{S}')$, and then compute the derivative via the expectation parameters $(\boldsymbol{\eta}, \mathbf{H})$. Unfortunately, converting between these three parameterizations requires $\mathcal{O}(M^3)$ computation. (To see why this is the case, note that computing $\mathbf{S}'$ essentially requires inverting the $\boldsymbol{\Theta}$ matrix.)

**A $\mathcal{O}(M^2)$ NGD update.** In what follows, we will demonstrate that the ELBO and its derivative can be computed from $\boldsymbol{\theta}$ and $\boldsymbol{\Theta}$ in $\mathcal{O}(M^2)$ time via careful bookkeeping. Consequently, NGD updates have the same asymptotic complexity as the other computations required for SVGP. Recall that the ELBO is given by

$$\text{ELBO} = \sum_{i=1}^{N} \overbrace{\underset{q(f(\mathbf{x}^{(i)}))}{\mathbb{E}} \left[ \log p(y^{(i)} \mid f(\mathbf{x}^{(i)})) \right]}^{\text{expected log likelihood}} - \text{KL}\left[ q(\mathbf{u}) \| p(\mathbf{u}) \right]$$

We will separately analyze the expected log likelihood and KL divergence computations.

## E.1   The Expected Log Likelihood and its Gradient

Assume we are estimating the ELBO from a single data point $\mathbf{x}, y$. The expected log likelihood term of the ELBO is typically computed via Gauss-Hermite quadrature or Monte Carlo integration [40]:[10]

$$\underset{q(f(\mathbf{x}))}{\mathbb{E}} \left[ \log p(y \mid f(\mathbf{x})) \right] = \sum_{s=1}^{S} w_s p(y \mid f_s), \quad f_s = \mu_{\text{aprx}}^*(\mathbf{x}) + \text{Var}_{\text{aprx}}^*(\mathbf{x})^{1/2} \varepsilon_s$$

where $w_s$ are the quadrature weights (or $1/S$ for MC integration) and $\varepsilon_s$ are the quadrature locations (or samples from $\mathcal{N}(0,1)$ for MC integration). Therefore, the variational parameters only interact with the expected log likelihood term via $\mu_{\text{aprx}}^*(\mathbf{x})$ and $\text{Var}_{\text{aprx}}^*(\mathbf{x})$. We can write its gradients via chain rule as:

$$\frac{\partial \mathbb{E}_{q(f(\mathbf{x}))}\left[\log p(y|f(\mathbf{x}))\right]}{\partial \boldsymbol{\eta}} = c_1 \frac{\partial \mu_{\text{aprx}}^*(\mathbf{x})}{\partial \boldsymbol{\eta}} + c_2 \frac{\partial \text{Var}_{\text{aprx}}^*(\mathbf{x})}{\partial \boldsymbol{\eta}}$$

$$\frac{\partial \mathbb{E}_{q(f(\mathbf{x}))}\left[\log p(y|f(\mathbf{x}))\right]}{\partial \mathbf{H}} = c_3 \frac{\partial \mu_{\text{aprx}}^*(\mathbf{x})}{\partial \mathbf{H}} + c_4 \frac{\partial \text{Var}_{\text{aprx}}^*(\mathbf{x})}{\partial \mathbf{H}} \tag{S16}$$

for some constants $c_1$, $c_2$, $c_3$, and $c_4$ that do not depend on the variational parameters. It thus suffices to show that the posterior mean/variance and their gradients can be computed from $\boldsymbol{\theta}$ and $\boldsymbol{\Theta}$ in $\mathcal{O}(M^2)$ time.

**The predictive distribution and its gradient.** All expensive computations involving $\boldsymbol{\theta}$ and $\boldsymbol{\Theta}$ are written in blue.

$\mu_{\text{aprx}}^*(\mathbf{x})$ and its derivative can be written as:

$$\mu_{\text{aprx}}^*(\mathbf{x}) = \mathbf{k}_{\mathbf{Zx}}^\top \mathbf{K}_{\mathbf{ZZ}}^{-1/2} \mathbf{m}' \qquad \text{(standard parameters)}$$

$$= \mathbf{k}_{\mathbf{Zx}}^\top \mathbf{K}_{\mathbf{ZZ}}^{-1/2} \boldsymbol{\eta} \qquad \text{(expectation parameters)}$$

$$= \mathbf{k}_{\mathbf{Zx}}^\top \mathbf{K}_{\mathbf{ZZ}}^{-1/2} (-2\boldsymbol{\Theta})^{-1} \boldsymbol{\theta}, \tag{S17}$$

$$\frac{\partial \mu_{\text{aprx}}^*(\mathbf{x})}{\partial \boldsymbol{\eta}} = \mathbf{K}_{\mathbf{ZZ}}^{-1/2} \mathbf{k}_{\mathbf{Zx}}, \tag{S18}$$

$$\frac{\partial \mu_{\text{aprx}}^*(\mathbf{x})}{\partial \mathbf{H}} = \mathbf{0}.$$

$\text{Var}_{\text{aprx}}^*(\mathbf{x})$ and its derivative can be written as:

$$\text{Var}_{\text{aprx}}^*(\mathbf{x}) = \mathbf{k}_{\mathbf{Zx}}^\top \mathbf{K}_{\mathbf{ZZ}}^{-1/2} \left(\mathbf{S}' - \mathbf{I}\right) \mathbf{K}_{\mathbf{ZZ}}^{-1/2} \mathbf{k}_{\mathbf{Zx}} \qquad \text{(standard parameters)}$$

$$= \mathbf{k}_{\mathbf{Zx}}^\top \mathbf{K}_{\mathbf{ZZ}}^{-1/2} \left(\mathbf{H} - \boldsymbol{\eta}\boldsymbol{\eta}^\top - \mathbf{I}\right) \mathbf{K}_{\mathbf{ZZ}}^{-1/2} \mathbf{k}_{\mathbf{Zx}} \qquad \text{(expectation parameters)}$$

$$= \mathbf{k}_{\mathbf{Zx}}^\top \mathbf{K}_{\mathbf{ZZ}}^{-1/2} \left((-2\boldsymbol{\Theta})^{-1} - \mathbf{I}\right) \mathbf{K}_{\mathbf{ZZ}}^{-1/2} \mathbf{k}_{\mathbf{Zx}}, \tag{S19}$$

$$\frac{\partial \text{Var}_{\text{aprx}}^*(\mathbf{x})}{\partial \boldsymbol{\eta}} = -2 \left(\mathbf{k}_{\mathbf{Zx}}^\top \mathbf{K}_{\mathbf{ZZ}}^{-1/2} (-2\boldsymbol{\Theta})^{-1} \boldsymbol{\theta}\right) \mathbf{K}_{\mathbf{ZZ}}^{-1/2} \mathbf{k}_{\mathbf{Zx}}, \tag{S20}$$

$$\frac{\partial \text{Var}_{\text{aprx}}^*(\mathbf{x})}{\partial \mathbf{H}} = \left(\mathbf{K}_{\mathbf{ZZ}}^{-1/2} \mathbf{k}_{\mathbf{Zx}}^\top\right) \left(\mathbf{k}_{\mathbf{Zx}}^\top \mathbf{K}_{\mathbf{ZZ}}^{-1/2}\right). \tag{S21}$$

In Eqs. (S17) to (S21), the only expensive operation involving $\mathbf{K}_{\mathbf{ZZ}}$ is $\mathbf{K}_{\mathbf{ZZ}}^{-1/2} \mathbf{k}_{\mathbf{Zx}}$, which can be computed with CIQ. The only expensive operation involving the variational parameters is $(-2\boldsymbol{\Theta})^{-1} \mathbf{K}_{\mathbf{ZZ}}^{-1/2} \mathbf{k}_{\mathbf{Zx}}$, which can be computed with preconditioned conjugate gradients after computing $\mathbf{K}_{\mathbf{ZZ}}^{-1/2} \mathbf{k}_{\mathbf{Zx}}$.[11] Those operations only need to be computed once, and then they can be reused across Eqs. (S17) to (S21). In total, the entire computation for the expected log likelihood and its derivative is $\mathcal{O}(M^2)$.

### E.2 The KL Divergence and its Gradient

We will demonstrate that the KL divergence and its gradient can be computed from $\boldsymbol{\theta}$ and $\boldsymbol{\Theta}$ in $\mathcal{O}(M^2)$ time. All expensive computations involving $\boldsymbol{\theta}$ and $\boldsymbol{\Theta}$ are written in blue.

The whitened KL divergence from Sec. 5.1 is given by:

$$\text{KL}\left[\,q(\mathbf{u}')\|p(\mathbf{u}')\,\right] = \frac{1}{2}\left[\mathbf{m}'^\top \mathbf{m}' + \text{Tr}\left(\mathbf{S}'\right) - \log|\mathbf{S}'| - M\right] \qquad \text{(standard parameters)}$$

$$= \frac{1}{2}\left[\text{Tr}\left(\mathbf{H}\right) - \log|\mathbf{H} - \boldsymbol{\eta}\boldsymbol{\eta}^\top| - M\right] \qquad \text{(expectation parameters)}$$

$$= \frac{1}{2}\left[\boldsymbol{\theta}^\top (-2\boldsymbol{\Theta})^{-2} \boldsymbol{\theta} + \text{Tr}\left((-2\boldsymbol{\Theta})^{-1}\right) + \log|-2\boldsymbol{\Theta}| - M\right]. \tag{S22}$$

The KL derivative with respect to $\boldsymbol{\eta}$ and $\mathbf{H}$ is surprisingly simple when re-written in terms of the natural parameters

$$
\begin{aligned}
\frac{\partial \operatorname{KL}\left[\,q(\mathbf{u}')\|p(\mathbf{u}')\,\right]}{\partial \boldsymbol{\eta}} &= \left(\mathbf{H} - \boldsymbol{\eta}\boldsymbol{\eta}^\top\right)^{-1}\boldsymbol{\eta} = (\mathbf{S}')^{-1}\boldsymbol{\eta} \\
&= \boldsymbol{\theta} \qquad\qquad\qquad\qquad\qquad\qquad\qquad\quad \text{(S23)} \\
\frac{\partial \operatorname{KL}\left[\,q(\mathbf{u}')\|p(\mathbf{u}')\,\right]}{\partial \mathbf{H}} &= \frac{1}{2}\mathbf{I} - \frac{1}{2}\left(\mathbf{H} - \boldsymbol{\eta}\boldsymbol{\eta}^\top\right)^{-1} = \frac{1}{2}\mathbf{I} - \frac{1}{2}(\mathbf{S}')^{-1} \\
&= \frac{1}{2}\mathbf{I} + \boldsymbol{\Theta}. \qquad\qquad\qquad\qquad\qquad\qquad \text{(S24)}
\end{aligned}
$$

Thus the derivative of the KL divergence only takes $\mathcal{O}(M^2)$ time to compute. The forward pass can also be computed in $\mathcal{O}(M^2)$ time—using stochastic trace estimation for the trace term [16, 29], stochastic Lanczos quadrature for the log determinant [20, 76], and CG for the solves. However, during training the forward pass can be omitted as only the gradient is needed for NGD steps.

## F  Experimental Details

**SVGP experiments.** Each dataset is randomly split into 75% training, 10% validation, and 15% testing sets; $\mathbf{x}$ and $y$ values are scaled to be zero mean and unit variance. All models use a constant mean and a Matérn 5/2 kernel, with lengthscales initialized to $0.01$ and inducing points initialized by $K$-means clustering. Each model is trained for 20 epochs with a minibatch size of $256$.[12] We alternate between optimizing $\mathbf{m}'/\mathbf{S}'$ and the other parameters, using NGD for the former and Adam [48] for the latter. Each optimizer uses an initial learning rate of $0.01$[13], decayed by $10\times$ at epochs 1, 5, 10, and 15. For CIQ we use $Q = 15$ quadrature points. msMINRES terminates when the $\mathbf{c}_j$ vectors achieve a relative norm of $0.001$ or after $J = 200$ iterations. We experimented with tighter tolerances and found no difference in the models' final accuracy. (Note that $J = 200$ is almost always enough to achieve the desired $0.001$ tolerance; see Fig. S7.) Results are averaged over three trials.

The 3DRoad [34] and CovType [9] datasets are available from the UCI repository [2]. For 3Droad, we only use the first two features—corresponding to latitude and longitude. For CovType, we reduce the 7-way classification problem to a binary problem (Cover_Type $\in \{2, 3\}$ versus Cover_Type $\in \{0, 1, 4, 5, 6\}$). The Precipitation dataset [52, 53] is available from the IRI/LDEO Climate Data Library.[14] This spatio-temporal dataset aims to predict the "WASP" index (Weighted Anomaly Standardized Precipitation) at various latitudes/longitudes. Each data point corresponds to the WASP index for a given year (between 2010 and 2019)—which is the average of monthly WASP indices. In total, there are 10 years and 10,127 latitude/longitude coordinates, for a total dataset size of 101,270.

**Bayesian optimization experiments.** The 6-dimensional Hartmann function is a classical test problem in global optimization[15]. There are 6 local minima and a global optimal value is $-3.32237$. We use a total of 100 evaluations with 10 initial points. The 10 initial points are generated using a Latin hypercube design and we use a batch size of 5. In each iteration, we draw 5 samples and select 5 new trials to evaluate in parallel.

We consider the same setup and controller as in [21] for the 12-dimensional Lunar Lander problem. The goal is to learn a controller that minimizes fuel consumption and distance to a given landing target while also preventing crashes. The state of the lunar lander is given by its angle and position, and their time derivatives. Given this state vector, the controller chooses one of the following four actions: $a \in \{$do nothing, booster left, booster right, booster down$\}$. The objective is the average final reward over a fixed constant set of 50 randomly generated terrains, initial positions, and initial velocities. The optimal controller achieves an average reward of $\approx 309$ over the 50 environments.

For both problems, we use a Matérn-5/2 kernel with ARD and a constant mean function. The domain is scaled to $[0, 1]^d$ and we standardize the function values before fitting the Gaussian process. The kernel hyperparameters are optimized using L-BFGS-B and we use the following bounds: (lengthscale) $\ell \in [0.01, 2.0]$, (signal variance) $s^2 \in [0.05, 50.0]$, (noise variance) $\sigma^2 \in [1e{-}6, 1e{-}2]$. Additionally, we place a horseshoe prior on the noise variance as recommended in [72]. We add $1e{-}4$ to the diagonal of the kernel matrix to improve the conditioning and use a preconditioner of rank 200 for CIQ.

**Image reconstruction experiments.** The matrix $\mathbf{A} = \boldsymbol{D}\boldsymbol{B}$ is given as the product of two matrices $\boldsymbol{D}$ and $\boldsymbol{B}$. Here $\boldsymbol{B}$ is a $N^2 \times N^2$ Gaussian blur matrix with a blur radius of 2.5 pixels and filter size of 5 pixels. The binary matrix $\boldsymbol{D}$ is a $KM^2 \times N^2$ downsampling or decimation matrix that connects the $N \times N$ high-resolution image to the $M \times M$ low-resolution images. For the hyperparameters $\gamma_{\mathrm{obs}}$ and $\gamma_{\mathrm{prior}}$ we choose Jeffrey's hyperpriors, i.e.

$$p(\gamma_{\mathrm{obs}}) \propto \gamma_{\mathrm{obs}}^{-1} \qquad \text{and} \qquad p(\gamma_{\mathrm{prior}}) \propto \gamma_{\mathrm{prior}}^{-1} \qquad \text{(S25)}$$

In order to conduct the experiment we use the observation likelihood with $\gamma_{\mathrm{obs}} = 1$ to sample $K = 4$ low-resolution images $\mathbf{y}_{1:K}$ from the high-resolution image. The discrete Laplacian matrix $\mathbf{L}$ is defined by the following isotropic filter:

$$\mathbf{L}_{\mathrm{filter}} = \frac{1}{12} \begin{bmatrix} 1 & 2 & 1 \\ 2 & -12 & 2 \\ 1 & 2 & 1 \end{bmatrix} \qquad \text{(S26)}$$

For both $\mathbf{L}$ and $\boldsymbol{B}$ we implicitly use reflected (i.e. non-periodic) boundary conditions. We use a CG tolerance of $0.001$ and a maximum of $J = 400$ msMINRES iterations. We use a Jacobi preconditioner for CG. We draw 1000 samples from the Gibbs sampler and treat the first 200 samples as burn-in. The reconstructed image depicted in the main text is the (approximate) posterior mean. In the main text we provided the conditional posterior for the latent image $\mathbf{x}$. To complete the specification of the Gibbs sampler we also need the posterior conditionals for $\gamma_{\mathrm{obs}}$ and $\gamma_{\mathrm{prior}}$, both of which are given by gamma distributions:

$$p(\gamma_{\mathrm{obs}}|\mathbf{x}, \mathbf{y}_{1:K}) = \mathrm{Ga}(\gamma_{\mathrm{obs}}|\alpha = 1 + \tfrac{KM^2}{2}, \beta = 2/||\mathbf{y}_{1:K} - \mathbf{A}\mathbf{x}||^2)$$
$$p(\gamma_{\mathrm{prior}}|\mathbf{x}) = \mathrm{Ga}(\gamma_{\mathrm{prior}}|\alpha = 1 + \tfrac{N^2-1}{2}, \beta = 2/||\mathbf{L}\mathbf{x}||^2) \qquad \text{(S27)}$$

## G    Proof of Theorem 1

To prove the convergence result in Thm. 1, we first prove the following lemmas.

**Lemma 2.** *Let $\mathbf{K} \succ 0$ be symmetric positive definite and let shifts $t_1, \ldots, t_Q > 0$ be real-valued and positive. After $J$ iterations of msMINRES, all shifted solve residuals are bounded by:*

$$\left\| (\mathbf{K} + t_q\mathbf{I})\mathbf{c}_J^{(q)} - \mathbf{b} \right\|_2 \leq \left( \frac{\sqrt{\kappa(\mathbf{K} + t_q\mathbf{I})} - 1}{\sqrt{\kappa(\mathbf{K} + t_q\mathbf{I})} + 1} \right)^J \|\mathbf{b}\|_2 \leq \left( \frac{\sqrt{\kappa(\mathbf{K})} - 1}{\sqrt{\kappa(\mathbf{K})} + 1} \right)^J \|\mathbf{b}\|_2,$$

*where $\mathbf{b}$ is the vector to solve against, $\mathbf{c}_J^{(1)}, \ldots, \mathbf{c}^{(Q)}$ are the msMINRES outputs, and $\kappa(\mathbf{K})$ is the condition number of $\mathbf{K}$.*

*Proof.* The convergence proof uses a polynomial bound, which is the standard approach for Krylov algorithms. See [e.g. 65, 69, 75] for an analogous proof for the conjugate gradients method and [e.g. 32] for a treatment of MINRES applied to both positive definite and indefinite systems.

At iteration $J$, the msMINRES algorithm produces:

$$\mathbf{c}_J^{(q)} = \underset{\mathbf{c}^{(q)} \in \mathcal{K}_J(\mathbf{K}, \mathbf{b})}{\arg\min} \left[ \left\| (\mathbf{K} + t_q\mathbf{I})\mathbf{c}^{(q)} - \mathbf{b} \right\|_2 \right], \quad q = 1, \ldots Q, \qquad \text{(S28)}$$

where without loss of generality we assume $\mathbf{c}_0^{(q)} = \mathbf{0}$ for simplicity. Using the fact that Krylov subspaces are shift invariant, we immediately have that

$$\mathbf{c}_J^{(q)} = \underset{\mathbf{c}^{(q)} \in \mathcal{K}_J(\mathbf{K} + t_q\mathbf{I}, \mathbf{b})}{\arg\min} \left[ \left\| (\mathbf{K} + t_q\mathbf{I})\mathbf{c}^{(q)} - \mathbf{b} \right\|_2 \right], \quad q = 1, \ldots Q. \qquad \text{(S29)}$$

Since $(\mathbf{K}+t_q\mathbf{I}) \succ 0$ we may invoke a result on MINRES error bounds for symmetric positive definite matrices [32, Chapter 3] to conclude that

$$\left\|(\mathbf{K} + t_q\mathbf{I})\mathbf{c}_J^{(q)} - \mathbf{b}\right\|_2 \leq \left(\frac{\sqrt{\kappa(\mathbf{K} + t_q\mathbf{I})} - 1}{\sqrt{\kappa(\mathbf{K} + t_q\mathbf{I})} + 1}\right)^J \|\mathbf{b}\|_2.$$

Observing that $\kappa(\mathbf{K} + t_q\mathbf{I}) \geq \kappa(\mathbf{K})$ for all $q$ since $t_q > 0$ concludes the proof.

$\square$

Lemma 2 is a very loose bound, as it doesn't assume anything about the spectrum of $\mathbf{K}$ (which is standard for generic Krylov method error bounds) and upper bounds the residual error for every shift using the most ill-conditioned system. In practice, we find that smMINRES converges for many covariance matrices with $J \approx 100$, even when the conditioning is on the order of $\kappa(\mathbf{K}) \approx 10^4$ and this convergence can be further improved with preconditioning.

**Lemma 3.** *For any positive definite $\mathbf{K}$ and positive $t$, we have*

$$\frac{\sqrt{\kappa(\mathbf{K} + t\mathbf{I})} - 1}{\sqrt{\kappa(\mathbf{K} + t\mathbf{I})} + 1} = \frac{\sqrt{\lambda_{max} + t} - \sqrt{\lambda_{min} + t}}{\sqrt{\lambda_{max} + t} + \sqrt{\lambda_{min} + t}} < \frac{\lambda_{max}}{4t} \tag{S30}$$

*Proof.* We can upper bound the numerator

$$\sqrt{\lambda_{\max} + t} - \sqrt{\lambda_{\min} + t} \leq \sqrt{\lambda_{\max} + t} - \sqrt{t}$$
$$= \sqrt{\lambda_{\max}}\left(\sqrt{1 + t/\lambda_{\max}} - \sqrt{t/\lambda_{\max}}\right) \leq \sqrt{\lambda_{\max}}\frac{1}{2\sqrt{t/\lambda_{\max}}} = \frac{\lambda_{\max}}{2\sqrt{t}}.$$

where we have applied the standard inequality $\sqrt{(\cdot) + 1} - \sqrt{(\cdot)} < \frac{1}{2\sqrt{(\cdot)}}$. The denominator can be (loosely) lower-bounded as $2\sqrt{t}$. Combining these two bounds completes the proof. $\square$

**Lemma 4.** *Let $\sigma_q^2$ and $\widetilde{w}_q$ be defined as in Eq. (S4). Then*

$$\sum_{q=1}^{Q} \frac{|w_q|}{|t_q|} = \sum_{q=1}^{Q} \frac{|\widetilde{w}_q|}{|\sigma_q^2|} < \frac{4Q \log\left(5\sqrt{\kappa(\mathbf{K})}\right)}{\pi\sqrt{\lambda_{min}}}$$

*where $w_q = -\widetilde{w}_q$ and $t_q = -\sigma_q^2$ as used in Eq. (S5).*

*Proof.* Using facts about elliptical integrals we have

$$\mathcal{K}'(k) < \log(1 + 4/k) \leq \log(5/k) \qquad k \in (0, 1) \qquad \text{([62, Thm. 1.7] and [84, Thm. 2])}$$
$$\frac{\pi}{2} \leq \mathcal{K}(k) \qquad k \in [0, 1] \qquad \text{([e.g. 62])}$$

where in the first statement we have used that $\mathcal{K}'(k) = \mathcal{K}(k')$. For Jacobi elliptic functions we have that

$$0 < \mathrm{dn}(u\mathcal{K}(k)|k) < 1 \qquad u \in (0, 1),\ k \in (0, 1) \qquad \text{([e.g. 56])}$$
$$0 < \mathrm{sn}(u\mathcal{K}(k)|k) < 1 \qquad u \in (0, 1),\ k \in (0, 1) \qquad \text{([e.g. 56])}$$
$$\mathrm{sn}(\pi u/2|0) < \mathrm{sn}(u\mathcal{K}(k)|k) < 1 \qquad u \in (0, 1),\ k \in (0, 1) \qquad \text{([10, Thm. 1])}$$

where in the last inequality we have used that $\mathcal{K}(0) = \pi/2$ [e.g. 1]. Coupling the final inequality above with $\mathrm{sn}(\pi u/2|0) = \sin(\pi u/2)$ for $u \in (0, 1)$ we have that

$$\sin(\pi u/2) < \mathrm{sn}(u\mathcal{K}(k)|k) < 1 \qquad u \in (0, 1),\ k \in (0, 1).$$

Now, for each $q$ we have that

$$\frac{w_q}{t_q} = \frac{\widetilde{w}_q}{\sigma_q^2} = \left(\frac{-2\sqrt{\lambda_{\min}}}{\pi Q \lambda_{\min}}\right)\frac{\mathcal{K}'(k)\mathrm{cn}\left(iu_q\mathcal{K}'(k) \mid k\right)\mathrm{dn}\left(iu_q\mathcal{K}'(k) \mid k\right)}{\mathrm{sn}(iu_q\mathcal{K}'(k) \mid k)^2}$$
$$= \left(\frac{2\mathcal{K}'(k)}{\pi Q\lambda_{\min}}\right)\frac{\mathrm{dn}\left(u_q\mathcal{K}(k') \mid k'\right)}{\mathrm{sn}(u_q\mathcal{K}(k') \mid k')^2} \qquad \text{(via Jacobi imaginary transforms [e.g. 1])}$$

Consequently, we may conclude that

$$\frac{|w_q|}{|t_q|} = \left(\frac{2\mathcal{K}'(k)}{\pi Q \lambda_{\min}}\right) \frac{\mathrm{dn}\,(u_q \mathcal{K}(k') \mid k')}{\mathrm{sn}(u_q \mathcal{K}(k') \mid k')^2}$$
$$\leq \frac{2 \log(5/k)}{\pi Q \lambda_{\min}} \left(\frac{1}{\sin^2(\pi u_q/2)}\right)$$

where we note that all quantities on the right hand side are positive. Plugging in the values of $k = 1/\sqrt{\kappa(\mathbf{K})}$, $u_q = (q-1/2)/Q$ and summing over $u_q$ we see that

$$\sum_{q=1}^{Q} \frac{|w_q|}{|t_q|} < \sum_{q=1}^{Q} \frac{2 \log\left(5\sqrt{\kappa(\mathbf{K})}\right)}{\pi Q \sqrt{\lambda_{\min}} \sin^2(\frac{\pi(q-1/2)}{2Q})}. \qquad \text{(S31)}$$

Through trigonometric identities $\sum_{q=1}^{Q} 1/(Q \sin^2 \frac{\pi(q-1/2)}{2Q}) = 2Q$ and, therefore,

$$\sum_{q=1}^{Q} \frac{|w_q|}{|t_q|} < \frac{4Q \log\left(5\sqrt{\kappa(\mathbf{K})}\right)}{\pi \sqrt{\lambda_{\min}}}.$$

$\qquad \square$

With these lemmas we are now able to prove Theorem 1:

**Theorem 1 (Restated).** *Let $\mathbf{K} \succ 0$ and $\mathbf{b}$ be inputs to msMINRES-CIQ, producing $\mathbf{a}_J \approx \mathbf{K}^{1/2}\mathbf{b}$ after $J$ iterations with $Q$ quadrature points. The difference between $\mathbf{a}_J$ and $\mathbf{K}^{1/2}\mathbf{b}$ is bounded by:*

$$\left\|\mathbf{v}_J - \mathbf{K}^{\frac{1}{2}}\mathbf{b}\right\|_2 \leq \overbrace{\mathcal{O}\left(\exp\left(-\frac{2Q\pi^2}{\log \kappa(\mathbf{K})+3}\right)\right)}^{\text{Quadrature error}} + \overbrace{\frac{2Q \log\left(5\sqrt{\kappa(\mathbf{K})}\right)\kappa(\mathbf{K})\sqrt{\lambda_{min}}}{\pi}\left(\frac{\sqrt{\kappa(\mathbf{K})}-1}{\sqrt{\kappa(\mathbf{K})}+1}\right)^{J-1}}^{\text{msMINRES error}} \|\mathbf{b}\|_2.$$

*where $\lambda_{max}, \lambda_{min}$ are the max and min eigenvalues of $\mathbf{K}$, and $\kappa(\mathbf{K})$ is the condition number of $\mathbf{K}$.*

*Proof.* First we note that the msMINRES-CIQ solution $\mathbf{a}_J$ can be written as $\sum_{i=1} w_q \mathbf{c}_J^{(q)}$, where $\mathbf{c}_J^{(q)}$ is the $q^{\text{th}}$ shifted solve $\approx (t_q\mathbf{I} + \mathbf{K})^{-1}\mathbf{b}$ from msMINRES. Applying the triangle inequality we have:

$$\left\|\mathbf{a}_J - \mathbf{K}^{\frac{1}{2}}\mathbf{b}\right\|_2 = \left\| \overbrace{\sum_{q=1}^{Q} w_q \mathbf{c}_J^{(q)} - \left(\mathbf{K}\sum_{q=1}^{Q} w_q (t_q\mathbf{I} + \mathbf{K})^{-1}\right)\mathbf{b}}^{\text{msMINRES error}} \right.$$

$$\left. + \underbrace{\left(\mathbf{K}\sum_{q=1}^{Q} w_q (t_q\mathbf{I} + \mathbf{K})^{-1}\right)\mathbf{b} - \mathbf{K}^{\frac{1}{2}}\mathbf{b}}_{\text{Quadrature error}} \right\|_2$$

$$\leq \sum_{q=1}^{Q} |w_q| \left\|\mathbf{c}_J^{(q)} - \mathbf{K}(t_q\mathbf{I} + \mathbf{K})^{-1}\mathbf{b}\right\|_2$$

$$+ \left\|\mathbf{K}\left(\sum_{q=1}^{Q} w_q (t_q\mathbf{I} + \mathbf{K})^{-1}\right)\mathbf{b} - \mathbf{K}^{\frac{1}{2}}\mathbf{b}\right\|_2 \qquad \text{(S32)}$$

Plugging Lemma 2 into the msMINRES part of the bound bound, we have:

$$\sum_{q=1}^{Q} |w_q| \left( \frac{\sqrt{\kappa(\mathbf{K} + t_q \mathbf{I})} - 1}{\sqrt{\kappa(\mathbf{K} + t_q \mathbf{I})} + 1} \right)^{J} \|\mathbf{b}\|_2$$

$$\leq \sum_{q=1}^{Q} |w_q| \left( \frac{\sqrt{\kappa(\mathbf{K} + t_q \mathbf{I})} - 1}{\sqrt{\kappa(\mathbf{K} + t_q \mathbf{I})} + 1} \right) \left( \frac{\sqrt{\kappa(\mathbf{K})} - 1}{\sqrt{\kappa(\mathbf{K})} + 1} \right)^{J-1} \|\mathbf{b}\|_2 \qquad \text{(via Lemma 2)}$$

$$\leq \sum_{q=1}^{Q} |w_q| \left( \frac{\lambda_{\max}}{4 t_q} \right) \left( \frac{\sqrt{\kappa(\mathbf{K})} - 1}{\sqrt{\kappa(\mathbf{K})} + 1} \right)^{J-1} \|\mathbf{b}\|_2 \qquad \text{(via Lemma 3)}$$

$$\leq \frac{2Q \log\left( 5\sqrt{\kappa(\mathbf{K})} \right) \lambda_{\max}}{\pi \sqrt{\lambda_{\min}}} \left( \frac{\sqrt{\kappa(\mathbf{K})} - 1}{\sqrt{\kappa(\mathbf{K})} + 1} \right)^{J-1} \|\mathbf{b}\|_2 \qquad \text{(via Lemma 4)}$$

$$\leq \frac{2Q \log\left( 5\sqrt{\kappa(\mathbf{K})} \right) \sqrt{\lambda_{\min}} \kappa(\mathbf{K})}{\pi} \left( \frac{\sqrt{\kappa(\mathbf{K})} - 1}{\sqrt{\kappa(\mathbf{K})} + 1} \right)^{J-1} \|\mathbf{b}\|_2 .$$

Plugging this bound and Lemma 1 into Eq. (S32) completes the proof. □

We can also prove this simple corollary:

**Corollary 1.** *Let $\mathbf{K} \succ 0$ and $\mathbf{b}$ be the inputs to Alg. 1, producing the output $\mathbf{a}'_J \approx \mathbf{K}^{-1/2} \mathbf{b}$ after $J$ iterations with $Q$ quadrature points. The difference between $\mathbf{a}_J$ and $\mathbf{K}^{1/2} \mathbf{b}$ is bounded by:*

$$\left\| \mathbf{a}'_J - \mathbf{K}^{-\frac{1}{2}} \mathbf{b} \right\|_2 \leq \overbrace{\mathcal{O}\left( \frac{1}{\lambda_{min}} \exp\left( -\frac{2Q\pi^2}{\log \kappa(\mathbf{K}) + 3} \right) \right)}^{\text{Quadrature error}} + \overbrace{\frac{2Q \log\left( 5\sqrt{\kappa(\mathbf{K})} \right) \kappa(\mathbf{K})}{\sqrt{\lambda_{min}} \pi} \left( \frac{\sqrt{\kappa(\mathbf{K})} - 1}{\sqrt{\kappa(\mathbf{K})} + 1} \right)^{J-1} \|\mathbf{b}\|_2}^{\text{msMINRES error}} .$$

*where $\lambda_{max}, \lambda_{min}$ are the maximal and minimal eigenvalues of $\mathbf{K}$, and $\kappa(\mathbf{K})$ is the condition number of $\mathbf{K}$.*

*Proof.* Note that $\mathbf{a}'_J = \mathbf{K}^{-1} \mathbf{a}_J$, where $\mathbf{a}_J$ is the msMINRES-CIQ estimate of $\mathbf{K}^{1/2} \mathbf{b}$. Using the sub-multiplicative property of the induced matrix 2-norm we see that

$$\left\| \mathbf{a}'_J - \mathbf{K}^{-\frac{1}{2}} \mathbf{b} \right\|_2 \leq \left\| \mathbf{K}^{-1} \right\|_2 \left\| \mathbf{a}_J - \mathbf{K}^{\frac{1}{2}} \mathbf{b} \right\|_2 = \frac{1}{\lambda_{\min}} \left\| \mathbf{a}_J - \mathbf{K}^{\frac{1}{2}} \mathbf{b} \right\|_2 ,$$

where the final term is bounded by Thm. 1. □

## Footnotes

[9] Both datasets are originally from the UCI repository and can be downloaded from https://github.com/gpleiss/ciq_experiments/tree/main/svgp/data.

[10] It can also be computed analytically for Gaussian distributions [39]. The analytic form achieves the same derivative decomposition as in Eq. (S16) and so the following analysis will still apply.

[11] We typically apply a Jacobi preconditioner to these solves.

[12] The batch size is 512 on the Covtype dataset due to its larger size.

[13] On the Precipitation dataset, the initial learning rate is 0.005 for NGD stability with the Student-T likelihood.

[14] A processed version of the dataset is available at `https://github.com/gpleiss/ciq_experiments/tree/main/svgp/data`. Original source of data: `http://iridl.ldeo.columbia.edu/maproom/Global/Precipitation/WASP_Indices.html`.

[15] `https://www.sfu.ca/~ssurjano/hart6.html`