[Reviews · NeurIPS 2020]

Review 1

Summary and Contributions: The authors propose to use a certain Krylov subspace approach for solving linear systems of the form K^{1/2} z and K^{-1/2} z based on a certain contour integral representation, where K is a kernel matrix arising in Gaussian process regression, in order to solve larger systems than it is possible to Cholesky factorize. UPDATE: after discussion, my view on this paper has largely not changed.

Strengths: This is a good paper. The authors' iterative solver proposal makes sense and in principle should allow one to scale sparse Gaussian processes to settings where there are as many as 50,000 inducing points, which is an order of magnitude more than one can do with Cholesky factorizations. This also avoids memory bottlenecks from assembling large kernel matrices, by rewriting all necessary expressions as matrix-vector products which can be computed in a matrix-free manner. The authors' error analysis is rigorous, makes sense, and *crucially* makes it clear that these techniques introduce error. Even better, the authors quantify said error in a way that makes it understandable, by deriving bounds using techniques familiar to anyone who has studied numerical analysis. This is in stark contrast to prior work on iterative solvers for GPs, which has incorrectly claimed that they give exact Gaussian processes, which is simply false due to numerical error arising from ill-conditioned matrix-vector operations. I applaud the authors for presenting their work honestly and not following these embarrassing trends. Finally, the authors present a number of nice extras, such as computation of natural gradients which can speed up optimization, and some reasonable experiments with inducing point GPs. There are also examples in Bayesian optimization which might be interesting to practitioners in that area.

Weaknesses: The main downside of the method is that these ideas involve heavy low-level coding work which makes them a pain for practitioners to use. People are unlikely to start using these techniques unless they really need a GP with a lot of inducing points, or someone else writes GPflow code for the solvers which is so good that practitioners can just drop it in without thinking about what it does. The other downside is that techniques like this are completely classical in the numerical analysis and partial differential equations communities, where people have known about Krylov solvers for a long time, and people have moved on to more sophisticated techniques such as multi-grid methods [1] ages ago. What is not so classical here is working with systems of the form A^{-1/2} b instead of much more commonly studied A^{-1} b, because the former arise in GP models. The Gibbs sampling for image reconstruction example is a bit bizarre - I am not sure why anybody would use GPs for this due to the overwhelming empirical success of deep networks for image reconstruction tasks. Moreover, Gibbs sampling itself is largely an outdated technique, which is often outperformed by Hamiltonian Monte Carlo and other more sophisticated methods. This example really ought to be replaced with something else, and is perhaps the weakest part of the paper. [1] W. E. Principles of Multiscale Modeling. Cambridge University Press, 2011.

Correctness: I looked over the authors' error analysis, and all looks reasonable to me, though I did not check the lemmas and derived inequalities line by line. The basic idea to split error into different sources and analyze them separately is pretty standard in numerical analysis. Some of the ideas use estimates from complex analysis, but this area is so well-studied that pretty much all the expressions used are likely classical by now. The experiments likewise seem reasonable, but I am not an expert in Bayesian optimization, so I cannot comment too much about how good or bad the results are, or if there are other relevant baselines that are missing.

Clarity: This paper is well-written and describes most of the issues well. Notation is consistent, unambiguous, and largely used well. I am especially happy that the authors added interpretable labels to their error bound, which makes it easier to read, and that they did not hide the constants in the appendix where people have to go looking for them - this makes the paper read much better. Overall, I did not have much trouble understanding the work, so I think the authors did a reasonable job in their writing.

Relation to Prior Work: The idea to replace Cholesky factorizations with iterative solvers in order to scale to larger systems than can be successfully Cholesky-factorized has a long and detailed history in numerical analysis. Most of that literature focuses on the case A^{-1} b where A is a very large sparse matrix, motivated by solving linear partial differential equations via the finite element method. In comparison, systems of the form A^{1/2} b and A^{-1/2} b are less studied, and tend to be written about in papers rather than in the classical numerical analysis books. More work is needed to develop, deploy, and popularize such approaches. However, there are also some citations in the statistics literature the authors may be unaware of - I list these below. [1] D. P. Simpson, I. W. Turner, and A. N. Pettitt. Fast sampling from a Gaussian Markov random field using Krylov subspace approaches. [2] D. P. Simpson, I. W. Turner, C. M. Strickland and A. N. Pettitt. Scalable iterative methods for sampling from massive Gaussian random vectors.

Reproducibility: Yes

Additional Feedback: The font size on some of the plots is too small and should be made bigger. Please ensure that all fonts used in the plots are the same size of the caption, so that the paper is accessible and can be read by those without perfect vision. Please also use colorblind-friendly colors.


Review 2

Summary and Contributions: 3rd EDIT: With the notebook now attached to this submission, I am content with the quality of the empirical evaluation. 2nd EDIT: It appears that the results presented in this paper highly depend on the floating point precision. For double precision, the Cholesky decomposition fails less often whereas MINRES needs more iterations to converge. Since this aspect is neither explored nor even mentioned, I recommend to reject this paper. EDIT: In their rebuttal the authors just brushed over my concern that the number of iterations is limited to J=200. It appears this parameter is far more crucial than the authors are willing to admit and reproducing the experiments turned out to be difficult. I can not accept this paper in good consciousness. This submission addresses the problem of solving linear equation systems where the coefficient matrix is the root (or inverse root) of an s.p.d. matrix. The approach is described in Section 3: using Cauchy's integral formula, the problem is transformed into a quadrature problem. This way the linear system is approximated as a sum of linear systems involving just the inverse of the matrix plus different scalar shifts. Then the authors propose to use MINRES to solve these systems. Without preconditioner, all systems can be solved using the same matrix-vector multiplications. Furthermore, the authors analyze the convergence rate of their algorithm and show how to compute gradients efficiently. The empirical evaluation explores the usefulness of the approach for variational GP approximations, Bayesian optimization with Thompson sampling and an image reconstruction task using Gibbs sampling.

Strengths: Although the main idea is not novel, I think it is important to forward it to the community. I was unaware of the proposed quadrature approach.

Weaknesses: Empirical Evaluation ==================== This part is incomplete and I think this paper should get one more iteration before being accepted. All in all, the experimental evaluation is decent but I want a more thorough exploration with regard to the number of MINRES iterations J which has been fixed to at most 200. How sensitive are the results with regard to this choice? More specifically, the question is what happens if the cap is too small, e.g. J=10, 25, 50. How many MINRES iterations are affordable before the approach becomes slower than the standard Cholesky factorization? What if the cap J is set equal to the dataset size N? What about less smooth and non-stationary kernel functions? Furthermore, I am missing a comparison to the related work that used conjugate gradients. For a resubmission it will be necessary to compare to https://arxiv.org/pdf/2002.09309.pdf which has been accepted at ICML.

Correctness: I skimmed over the proof of Theorem 1 in the appendix. The proofs of Lemma 2 and 3 seem reasonable. Lemma 4 is not in my area of expertise and I hope that the other reviewers have more to say about it. For the authors: is K(k) introduced somewhere? I couldn't find it.

Clarity: Nothing to complain

Relation to Prior Work: The discussion of related work is short but it appears to be sufficient.

Reproducibility: No

Additional Feedback: The experiments reported in Figure 1 are not reproducable. Please add the code to produce these figures or describe precisely how the matrices where generated. l. 19: The sentence about the N~5000 limitation is debatable. Why not remove it. l. 64: maybe add: (in theory) exact after N iterations l. 75: please augment references to [37] with a page reference


Review 3

Summary and Contributions: The aim is faster matrix computations of the form L^{-1} b where L is the Cholesky decomposition of an N x N covariance matrix and b is a RHS vector. The authors are correct that Cholsky methods are limited to about N = 5,000 as applications tend to require many iterations. They achieve the speed-up by Krylov subspace methods, and the matrix-vector operations are amenable to fast GPU implementation. The method is applied to three important statistical / machine learning tools.

Strengths: The applications illustrate the wide range of important problems that can be effectively tackled with these methods. Throughout, but especially in the discussion, a balanced view of the advantages and disadvantages of the methodology is offered.

Weaknesses: The O(N^3) to O(N^2) advantage is weakened in practice by the small multiplicative constant for Cholesky versus a larger (but apparently moderate) constant for the Krylov subspace methods. The advantage is also dissipated when there are many RHS vectors b, a limitation acknowledged by the authors. In Fig. 1 the relative error appears to level off as Q increases. Would you not expect larger Q to continue to improve the error? Does this imply you cannot achieve high accuracy (versus good enough) without huge Q? In Section 5.2 search for a new acquisition point is done by brute force over a large number of random points. But one doesn't want to replace an expensive optimization with another. Some methods, e.g. Jones et al, Journal of Global Optimization 1998, exploit bounds on the GP posterior mean and variance in the search. Can those ideas be applied here?

Correctness: The method achieves "4 or 5 decimal places of accuracy" (line 40). This may be enough for the variational Bayes and BO applications where progress rather than exactness is enough. But what about say posterior predictive means or variances computed from these quantities? Are they still to the same accuracy? I didn't understand the claim about storage being reduced from O(N^2) (line 71). A Cholesky composition can overwrite the covariance matrix K with O(N) workspace too, I believe. Do the authors mean that computation of the N x N matrix K can be avoided?

Clarity: Overall, the paper is extremely well-written. The intuition of Krylov Subspace Methods implemented via matrix-vector multiplication (MVM)) is well done. There are plenty of references for the details, and the paper doesn't get too bogged down in the matrix computations. I didn't find any typos or minor grammatical errors. At line 61 it would help to have a bit more explanation why L b is useful (as opposed to L^{-1} b, which is clearly of utility).

Relation to Prior Work: Thorougly done.

Reproducibility: Yes

Additional Feedback:


Review 4

Summary and Contributions: This paper provides a new approach to approximating the action of matrix square roots (and inverse square roots) on vectors. A “contour integral quadrature” approach is adopted, in which the required square root is computed using a classical integral formulation that is then approximated using quadrature. This yields a number of linear systems that must be solved, which are tackled using a Krylov-based solver that is carefully constructed to exploit the relationship between the problems to reduce computational cost. The result is an algorithm for computing square roots, and inverse square roots, that has the distinct advantages of (1) only requiring matrix-vector multiplications (2) can more easily be GPU accelerated (3) has minimal memory footprint.

Strengths: This is a strong paper in my view. The importance of matrix square roots and inverse square roots to the NeurIPS community (and, indeed, much of the computational mathematics community) is clear and well-motivated in the paper. The discussion of how to obtain gradients for the procedure makes this work particularly well-suited to the NeurIPS community. The theoretical results provided are natural and what one would expect to see for such a paper. The experimental validation in Sections 4 and 5 feels complete, on the whole (with a few minor gaps that I discuss under “weaknesses”). In particular, the applications in Section 5 come across as both challenging and realistic, and the performance exhibited by the proposed method is very impressive.

Weaknesses: Krylov methods often suffer from a high degree of numerical instability which was not mentioned or explored in the paper or its supplement. I believe this warrants mention, and ideally some exploration in the text, though I appreciate that there may not be space in the paper for a detailed exploration. One might well start to see issues with convergence if the tolerance were set to below the stated value of 10^-4, and readers should at least be informed of this even if the impact of it is not explored. The example detailed in Section 5.3 is impressive but feels rather incomplete - a comparison to a classical method for computing the Cholesky would help to provide context here. I expect that the point the authors are trying to make here is that 25k x 25k problem that is tackled is not accessible to classical Cholesky factorisations, but a plot of wall time per sample vs. dimension, for both the new solver and a classical Cholesky solver would help the reader to ascertain how impressive 0.61s per-sample really is.

Correctness: While the algorithm certainly appears to work in practise, I do not believe that Theorem 1 is correct. The proof relies on Lemma 2 in the supplement, which claims that the following bound holds: ( (\sqrt{\kappa(K + t I)} - 1) / (\sqrt{\kappa(K + t I)} + 1 )^J \leq ( (\sqrt{\kappa(K)} - 1) / (\sqrt{\kappa(K)} + 1) )^J This is clearly not the case. For a trivial counterexample set K = 4 * I and t = 5 - then on the left we have (1/2)^J and on the right (1/3)^J - the former is clearly larger than the latter for any positive J. While this is a small error and does not dismiss the approach (clearly the method works by examining the experiments), unfortunately it renders Theorem 1 incorrect and is probably too large an error to correct at this stage. Property 1 appears to claim that the resulting algorithm has a computational cost independent of the number of quadrature points Q. This appears only to represent the cost of msMINRES. Examining Appendix B one can see that the cost from CIQ must be linear in Q with a multiplier that depends on how \lambda_min and \lambda_max are estimated (if the method in B.3 is used, this is likely also \xi(K)). While I appreciate that this is negligible, I would suggest that the authors modify the text slightly to address this - either by explicitly including the dependence on Q or by noting that it is negligible when Q is small. EDIT: thanks to the authors for addressing this concern completely, and apologies for my mistake in assessing your proofs!

Clarity: The paper is generally extremely well written, clear and easy to follow.

Relation to Prior Work: A good set of citations is provided. The authors highlight, several “competing” methods for this problem (both existing Krylov methods in sec 3.4 and other approaches to rank reduction in sec 5.2) and discuss their relative merits. It would have been nice to see a more concrete comparison to the authors’ cited works [3, 4, 12].

Reproducibility: Yes

Additional Feedback: On line 183 there is an unfortunately placed footnote which makes the computational cost appear to be O(M^3)^3 - this might be wise to move! It is unfortunate that Theorem 1 is incorrect, as I very much like this paper.

[Author Response · NeurIPS 2020]

We thank the reviewers for their helpful feedback, and we hope to address your remarks here and in the revision. We
note that R4 questioned the correctness of Thm. 1 (specifically, the claim made in Lemma 2). While we thank R4 for a
careful review of Appx. G, we are *strongly* convinced that Thm. 1 is correct and hope to clarify the misunderstanding.

**Correctness of Thm. 1 (R4).** Your question revolves around a specific inequality in the proof of Lemma 2:

$$\left(\frac{\sqrt{\kappa(\mathbf{K}+t_q\mathbf{I})}-1}{\sqrt{\kappa(\mathbf{K}+t_q\mathbf{I})}+1}\right)^J \|\mathbf{b}\|_2 \leq \left(\frac{\sqrt{\kappa(\mathbf{K})}-1}{\sqrt{\kappa(\mathbf{K})}+1}\right)^J \|\mathbf{b}\|_2. \tag{1}$$

*Your counter example.* "Set $\mathbf{K} = 4\mathbf{I}$ and $t = 5$—then on the left we have $(1/2)^J$ and on the right $(1/3)^J$—the former is
clearly larger." We believe you may accidentally used the max eigenvalue alone in these calculations, rather than $\kappa(\mathbf{K})$.

For symmetric positive definite $\mathbf{K}$, the condition number $\kappa(\mathbf{K}) \triangleq \lambda_{\max}/\lambda_{\min}$ (max and min eigenvalues). Adding $t\mathbf{I}$
increases both the max and min eigenvalues; thus $\kappa(\mathbf{K} + t\mathbf{I}) = (\lambda_{\max} + t)/(\lambda_{\min} + t)$. Using your specific numbers,
we have $\kappa(\mathbf{K}) = 4/4 = 1$ and $\kappa(\mathbf{K} + t_q\mathbf{I}) = 9/9 = 1$. Plugging these into Eq. (1) we have that both sides equal 0.

*Proof.* Here we carefully show that Eq. (1) holds for any symmetric positive definite matrix $\mathbf{K}$ and $t \geq 0$. Note that
$\kappa(\mathbf{K} + t\mathbf{I}) = \frac{\lambda_{\max}+t}{\lambda_{\min}+t} \leq \frac{\lambda_{\max}}{\lambda_{\min}} = \kappa(\mathbf{K})$, which holds as long as $\lambda_{\max} \geq \lambda_{\min} > 0$ (true here as $\mathbf{K} \succ 0$). From here,
note that $\frac{a-1}{a+1} \leq \frac{b-1}{b+1} < 1$ whenever $0 \leq a \leq b$. Setting $a = \sqrt{\kappa(\mathbf{K} + t_q\mathbf{I})}$ and $b = \sqrt{\kappa(\mathbf{K})}$ and noting that condition
numbers are always at least 1 (implying the square root preserves the ordering), we have $a \leq b$, and thus Eq. (1) holds.

**"Krylov methods often suffer from a high degree of numerical instability" (R4).** Our method has two key
advantages that improve stability. First, we only use Krylov methods to solve linear systems rather than eigenvalue
problems. Common numerical pitfalls that hinder Krylov eigen-solvers (e.g. loss of orthogonality between Lanczos
vectors) have been shown to have little empirical effect on linear system solvers like MINRES and CG [e.g. Trefethen
and Bau, 1997; Fong and Saunders, SQUJS 2012]. Second, each solve is inherently a shifted system $\mathbf{K} + t_q\mathbf{I}$. While
Thm. 1 is in terms of the conditioning of $\mathbf{K}$ (because $\min_q t_q \to 0$ as $Q \to \infty$), in practice these shifts *dramatically*
improve the conditioning of $\mathbf{K}$, with $\min_q t_q \geq 1\mathrm{e}{-3}$ when $Q < 20$. This allows us to work directly with the matrix $\mathbf{K}$
rather than having to first add diagonal jitter. We will discuss this more in the revision.

**Accuracy of square roots (R3).** (*"In Fig. 1 the relative error appears to level off as $Q$ increases."*) For these
experiments, we stopped msMINRES at a tolerance of $1\mathrm{e}{-4}$, as we viewed $0.01\%$ error as sufficient for most tasks.
Consequentially, as $Q$ increases the error converges to the solver tolerance. (*Is 4 or 5 decimal places enough for*
*predictive means/variances?*) We believe this is often sufficient; we tried tighter tolerances and found no difference.

**Running time and storage (R3, R4).** **R3:** (*"I didn't understand. . . storage being reduced from $\mathcal{O}(N^2)$."*) Using our
method, we can avoid storing the $\mathcal{O}(N^2)$ kernel matrix if the MVMs are computed in a map-reduce fashion. While the
Cholesky factorization can be performed in-place, the artifact it produces still requires $\mathcal{O}(N^2)$ storage. We will clarify
this in the revision. **R4:** (*"The cost from CIQ must be linear in $Q$."*) We agree that the quadrature running time depends
on $Q$, but view this as negligible since it crucially does not impact the number of MVMs performed with $\mathbf{K}$. Thank you
for pointing this out; we will clarify this.

**Missing citations (R1).** Thank you, we will add these citations. While many communities have extensively studied
Krylov methods, as you note—the $\mathbf{K}^{-1/2}\mathbf{b}$ problem we are interested in has received far less attention than Krylov-
based matrix solves. We would again highlight our novel contributions in this space: a simple vector recurrence for
$\mathbf{K}^{-1/2}\mathbf{b}$, a detailed error analysis, efficient NGD updates, a simple backward pass, and a mechanism for preconditioning
multiple shifted solves up to an orthogonal transformation.

**Comparisons (R2).** (*"It will be necessary to compare to [Wilson et. al, ICML 2020]."*) This was not published as
of the NeurIPS submission, as you point out. Wilson et al. use RFFs to sample from the prior and an inducing point
approximation of the conditional to convert prior samples into posterior samples. Our method could augment their
approach, allowing for more inducing points and/or replacing RFFs for prior sampling. (*"Missing a comparison to the*
*related work that used CG."*) We would argue that the pros/cons of CG versus MINRES has been studied exhaustively
(e.g. [Fong and Saunders, SQUJS 2012]). Our use of MINRES enables the proof of our main convergence result. CG's
error bound uses a different norm and cannot be easily combined with the "quadrature error" term in Thm. 1.

**Empirical evaluation (R2).** (*"The number of MINRES iterations $J$ has been fixed to 200."*) The stopping criteria is a
specified residual tolerance ($10^{-3}$ or $10^{-4}$) or 200 iterations, whichever comes first. In practice the tolerance is almost
always met before $J = 200$; we will note this in the revision. (*"How many MINRES iterations. . . before the approach*
*becomes slower than Cholesky?"*) This depends on matrix size. For $N = 5{,}000$ it would take approximately $J = 1{,}000$
iterations, after which CIQ has more than converged.

[Meta-Review · NeurIPS 2020]

After multiple discussions with regards to the experimental evaluation and the use of a hard coded limit J=200 for MINRES, all the reviewers reached a consensus on accepting this paper as they believe it opens up a different direction for scalable GPs and related areas and it provides interesting theoretical analyses. I agree with the reviewers and recommend acceptance. I strongly recommend the authors discuss this thoroughly in their final version along with the additional material presented to the AC via CMT.